# Infrared driven hot electron generation and transfer from non-noble metal plasmonic nanocrystals

Dongming Zhou[1], Xufeng Li[1], Qiaohui Zhou[1] & Haiming Zhu [1,2✉]

Non-noble metal plasmonic materials, e.g. doped semiconductor nanocrystals, compared to their noble metal counterparts, have shown unique advantages, including broadly tunable plasmon frequency (from visible to infrared) and rich surface chemistry. However, the fate and harvesting of hot electrons from these non-noble metal plasmons have been much less explored. Here we report plasmon driven hot electron generation and transfer from plasmonic metal oxide nanocrystals to surface adsorbed molecules by ultrafast transient absorption spectroscopy. We show unambiguously that under infrared light excitation, hot electron transfers in ultrafast timescale (<50 fs) with an efficiency of 1.4%. The excitation wavelength and fluence dependent study indicates that hot electron transfers right after Landau damping before electron thermalization. We revealed the efficiency-limiting factors and provided improvement strategies. This study paves the way for designing efficient infrared light absorption and photochemical conversion applications based on non-noble metal plasmonic materials.

[1] The Centre for Chemistry of High-Performance & Novel Materials, Department of Chemistry, Zhejiang University, Hangzhou, Zhejiang 310027, China. [2] State Key Laboratory of Modern Optical Instrumentation, Zhejiang University, Hangzhou, Zhejiang 310027, China. ✉email: hmzhu@zju.edu.cn

Surface plasmons are charge-density oscillations at the surface of a conducting material and decay dominantly by creating energetic ("hot") electrons when it comes to nano size, known as Landau damping[1–3]. The strong light absorption and hot electron/hole harvesting from plasmonic metal nanostructures have been extensively reported, where plasmon induced hot electron transfer (PIHET) occurs at interface for photochemical[4–8], photovoltaic[9,10], and photodetection applications[11,12].

While PIHET from noble metal nanostructures continue to be an active research area, degenerately doped semiconductor nanocrystals (NCs) also exhibit plasmon features with broadly tunable frequency from visible to mid-IR[13]. And unlike metal nanostructures with relatively inert surface, doped semiconductor NCs with rich coordination surface sites enable feasible and tunable surface modification[14]. Despite these unique properties, however, hot-electron harvesting and conversion from doped semiconductor-based plasmon has been rarely reported, except a few recent studies of plasmon induced charge transfer in semiconductor nano-heterostructures and their applications in photocatalysis[15–17] and photodetections[18,19]. Plasmon-driven charge injection from doped semiconductors to molecules remains to be established.

For efficient PIHET, the interfacial electron transfer should be comparable with the ultrafast energy relaxation through electron–electron (e–e) and electron–phonon (e–ph) scattering[20]. Previous studies on metal nanostructures show that PIHET can occur by indirect or direct charge transfer. The former assumes that the hot electrons generated in metal nanostructures with energy higher than the lowest-unoccupied-molecular-orbital (LUMO) of the acceptor molecules or metal-semiconductor Schottky barrier can inject into molecules or semiconductors[21]. In direct transfer mechanism, electron transfer is triggered by direct excitation and dephasing of plasmonic field[22,23], which often requires direct bonding or state mixing at interface[24]. To understand and optimize PIHET process, one must have a direct and simultaneous view on the dynamics of excitation, decay, and transfer of plasmon induced hot electrons in both time and energy domain[2,22–25].

Here, we use F and In co-doped CdO (FICO) NCs as a model plasmonic semiconductor NC and Rhodamine B (RhB) as electron accepting molecule and investigate ultrafast generation and relaxation of hot electrons in plasmonic semiconductor NCs and hot-electron injection to molecular acceptors. Using transient absorption (TA) spectroscopy, we follow hot-electron distribution in FICO and transfer to RhB. Together with the pump wavelength- and power-dependent studies, we demonstrate and provide a mechanistic picture of indirect hot-electron transfer from plasmonic semiconductor NCs after Landau damping but before electron thermalization.

## Results

### Sample preparation and characterization for FICO NCs.
FICO NCs were synthesized by colloidal method according to the procedure developed by Ye et al.[26]. The synthesized NC has a plasmon peak at around 1750 nm (Fig. 1a) and an averaged diameter of 11.3 nm (Fig. 1b). From the peak position, we estimated the doping electron density of $\sim 1.1 \times 10^{21}$ cm$^{-3}$ (Supplementary Note 1, Supplementary Table 1, and Supplementary Fig. 1). We choose RhB as electron acceptor since it has an appropriate LUMO position (or reduction potential) of $-3.94$ V (vs vacuum)[27] and can bind to metal oxide surface through carboxylic group[28]. We prepared FICO–RhB complex by sonicating FICO hexane solution with added RhB and subsequently filtering the unadsorbed RhB since RhB molecule itself is not soluble in hexane. We estimated the average number of RhB

molecules per FICO NC to be ~90 based on their extinction coefficients (Supplementary Note 2). After RhB adsorption, the plasmon peak of FICO is red-shifted and broadened slightly (<10%, Fig. 1a and Supplementary Note 1), which is likely due to the combined effect of partial aggregation of NCs[29] and electronic coupling between FICO and adsorbed RhB[30]. The absence of significant broadening of FICO plasmon resonance and RhB absorption (Supplementary Fig. 2) precludes the strong hybridization between FICO and RhB[21].

After absorbing light, plasmon resonance damps its energy to hot electrons around Fermi level ($E_f$)[2], therefore determining $E_f$ is a prerequisite. $E_f$ in degenerately doped semiconductors sits in conduction band and shifts depending on doped carrier density. Here we estimated $E_f$ ($-4.62$ V vs vacuum) based on the valence band position of undoped CdO ($E_v = -8.11$ V vs vacuum)[31] and the optical bandgap ($B_{g,opt} = 3.49$ eV) in doped CdO from the Tauc plot (Fig. 1c) by $E_f = E_v + B_{g,opt}$ (Fig. 1d). There is an absorption tail extending to ~0.3 eV below $B_{g,opt}$, which is likely due to contributions from thermal broadening of electron distribution and sample heterogeneity. We note $B_{g,opt}$ in degenerately doped semiconductors has two opposite contributions, Burstein–Moss shift due to band filling and band renormalization due to electron–electron and electron–ion Columbic interaction[31,32]. The band renormalization is estimated to be about 0.3 eV based on doping electron density[33]. With direct bandgap energy of undoped CdO (2.16 eV)[34], the bandgap $B_g'$ after band renormalization is 1.86 eV. Therefore, the conduction band minimum ($E_{CBM}$) and $E_f$ locate at $-6.25$ and $-4.62$ V (vs vacuum), respectively, that is, $E_f$ is 1.63 V higher than $E_{CBM}$ (Fig. 1d).

### Hot-electron generation and relaxation in FICO.
We first use TA spectroscopy to follow photoexcitation dynamics of FICO NCs. FICO NCs were pumped by 1650-nm laser pulse (instrument response function or IRF of 200 fs, pump fluence of 36.5 μJ cm$^{-2}$) and probed with a visible (400–700 nm) white light continuum (see Methods for details). The two-dimensional plot of TA spectra of FICO is shown in Supplementary Fig. 3 with a few representative TA spectra at indicated delay times in Fig. 2a. Generally, FICO NCs show a broad induced absorption (IA) feature between 670 and 400 nm. Interestingly, the onset of IA spectra initially starts at ~450 nm at $-0.05$ ps and progressively shifts to ~670 nm in 0.3 ps (Fig. 2a) and remains there afterward (Supplementary Figs. 3 and 4). The 670 nm onset matches well with bandgap $B_g'$ (1.86 eV) (Fig. 2c right panel). As we will discuss below, the shifting of IA onset directly reflects the hot-electron thermalization process.

It is known that right after photoexcitation, plasmon resonance damps its energy and excites electrons from the states near $E_f$ thus depopulates those states in <10 fs through Landau damping[3,35]. Considering the conservation of photon energy (1650 nm or 0.75 eV)[35] and the relaxed momentum conservation in NCs[36], the lowest energy electron that can be excited is 0.75 eV below $E_f$. This process should generate an interband IA starting from ~2.74 eV ($=3.49-0.75$) or 453 nm (Fig. 2c left panel). This predicted value agrees very well with ~450 nm IA onset at $-0.05$ ps observed experimentally, confirming the proposed picture (Fig. 2c left panel). Afterward, electron thermalization occurs through e–e and e–ph scattering until a Fermi–Dirac distribution forms at 0.3 ps (Fig. 2c right panel). Finally, the hot-electron distribution cools by e–ph scattering. We note FICO has a rocksalt (cubic) structure[26] with flat valence band[37], therefore the IA spectra directly reflects the electron distribution in conduction band (Fig. 1d). We modeled the IA spectra after electron thermalization (0.3 ps) to extract the electron temperature ($T_e$)

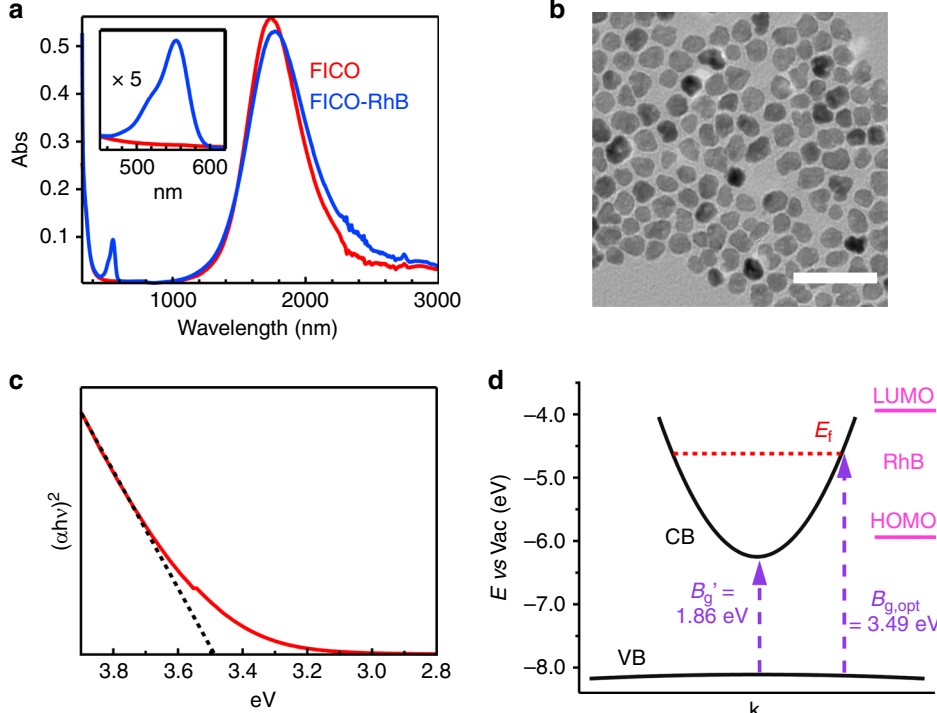

**Fig. 1 Characterization of FICO nanocrystals. a** Absorption of FICO nanocrystals (NCs) and FICO–Rhodamine B (RhB) complex. Insert shows the RhB absorption with vertical axis magnified by 5 times (×5). **b** TEM image of FICO NCs. **c** Tauc plot (red line) of FICO NCs and its liner fit (black dotted line) showing the optical bandgap of FICO at ~3.49 eV. **d** Band diagram of the FICO NCs and RhB. $E_f$ is Fermi level, $B_{g,opt}$ is the optical bandgap and $B'_g$ is the direct bandgap after band renormalization correction. The arrows show the optical transitions with energy of $B'_g$ and $B_{g,opt}$.

based on Fermi–Dirac distribution and parabolic band dispersion (Supplementary Note 3). The IA spectrum at 0.3 ps can be well described with $T_e$ of 4500 K (Fig. 2a). The theoretical maximum electron temperature $T_{e,max}$ assuming all absorbed photon energies are transferred to conduction band electrons without energy loss from phonon emission was calculated to be 5763 K (Supplementary note 4). The extracted lower $T_e$ for thermalized electrons indicates the onset of e–ph scattering during electron thermalization. A representative IA kinetics at 550 nm is shown in Fig. 2b and the rising (decay) process corresponds to electron thermalization (cooling) process. We fit the kinetics with an exponential rise and decay function convoluted with IRF, i.e., $\Delta T/T(t) = \text{IRF} \otimes \left(-e^{-t/\tau_r} + e^{-t/\tau_D}\right)$ where $\tau_r$ and $\tau_D$ is the rising and decay lifetime constant, respectively. IA reaches maximum at ~0.3 ps with an electron thermalization lifetime of 150 fs and decays to zero in ~1.3 ps with cooling lifetime of 0.21 ps.

**Hot-electron transfer from FICO**. The TA spectra of FICO–RhB complex with same FICO NC concentration and experiment conditions are shown in Fig. 3a (at selected delay times) and Supplementary Fig. 3 (full contour plot). We also show TA spectra of FICO at same delay time in Fig. 3a for comparison. Interestingly, the TA spectrum of FICO–RhB complex shows similar IA feature as that of FICO but is overlapped with a distinct bleach at ~560 nm (Fig. 3a). Subtracting 0.2 ps TA spectrum of FICO–RhB by that of FICO yields a difference spectrum in Fig. 3b. It shows a strong ground state bleach (GSB) peak matching exactly with RhB absorption and an IA peak at ~412 nm which can be attributed to RhB anion radical (Supplementary Fig. 5)[38]. We did not observe any shifting or broadening signature of RhB absorption on TA spectra, indicating negligible contribution from dielectric environment change or Stark effect due

to photoexcitation. This point will be further confirmed with excitation wavelength dependent study later. Because 1650 nm cannot excite RhB molecule and hole transfer is energetically unfavored, the simultaneous formation of RhB GSB and anion radical IA indicates PIHET from photoexcited FICO to surface adsorbed RhB molecules (Fig. 3c inset). Because RhB anion radical is weaker without known extinction coefficient and overlaps with strong IA signal from FICO, we use RhB GSB for following quantitative analysis on kinetics and signal size. The GSB kinetics of RhB is shown in Fig. 3c and we fit it with an exponential rise and decay function convoluted with IRF, i.e., $\Delta T/T(t) = \text{IRF} \otimes \left(-e^{-t/\tau_r} + e^{-t/\tau_D}\right)$. The lifetime of rise and decay process of RhB bleach kinetics are <50 fs (beyond time resolution) and 407 fs, respectively, corresponding to electron transfer to RhB and subsequent back electron transfer process. The much faster electron transfer to RhB than electron thermalization in FICO suggests PIHET can compete with electron thermalization, prevailing over energy loss from e–e and e–ph scattering.

As we introduced in the beginning, PIHET can be either direct or indirect. After photoexcitation under same experimental conditions, FICO and FICO–RhB show almost same IA spectra at wavelength range (>600 nm) where RhB does not absorb (Fig. 3a), suggesting little energy dissipated thus low electron transfer efficiency. We calculated the PIHET quantum yield (QY) by the ratio between reduced RhB and absorbed photon (Supplementary Note 5). With 36.5 μJ cm$^{-2}$, 1650-nm excitation, the efficiency is estimated to be 0.75%. Direct transfer through plasmon dephasing process usually manifests as high efficiency and significantly broadened plasmon resonance[22]. The low efficiency, together with only slightly (<10%) broadened plasmon peak, suggests PIHET in FICO–RhB likely occurs through indirect mechanism with hot-electron generation from Landau damping followed by transfer process. This will be further investigated and confirmed below.

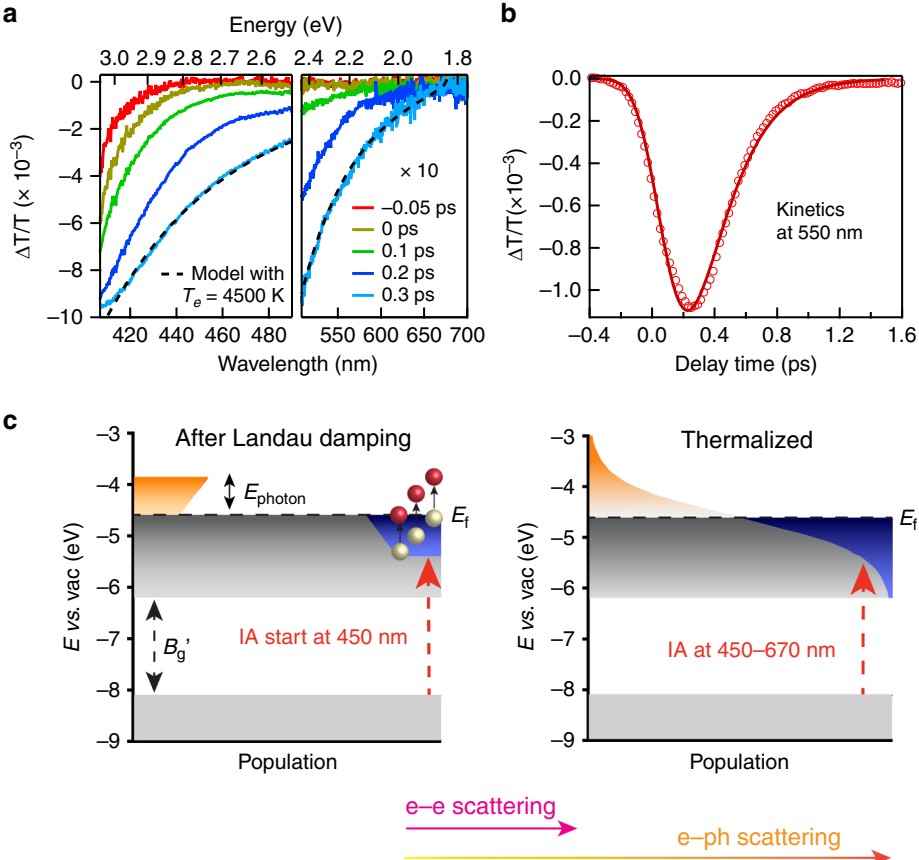

**Fig. 2 Hot-electron generation and relaxation in FICO nanocrystals. a** Transient absorption (TA) spectra of FICO nanocrystals (NCs) at different delay times with 400–490 nm in left panel and 500–700 nm in right panel. For better view, the amplitude of TA spectra of −0.05 and 0 ps has been multiplied by four times and the vertical axis in right panel (500–700 nm) has been zoomed in by 10 times (×10). Black dashed line shows modeled spectrum at 0.3 ps with $T_e$ of 4500 K. **b** TA kinetics of FICO NCs probed at 550 nm (empty circles) and the single-exponential fitting (red line) of the rise and decay process. **c** Scheme showing carrier distribution after hot-electron generation from Landau damping (left) and after electron thermalization through e–e and e–ph scattering (right). $E_{photon}$ is the pump photon energy. $E_f$ is the Fermi level. $B'_g$ is the direct bandgap after band renormalization correction. Red dashed arrows show the induced absorption (IA) process. The gray region shows the undisturbed filled states. The orange and blue regions represent the excited electrons and empty states left, respectively, after Landau damping (left) and thermalization (right).

To confirm the proposed scenario that PIHET occurs through indirect mechanism but before electron thermalization, we first performed pump fluence dependent study. The reduced RhB shows a linear dependence on pump fluence (Fig. 4a), corresponding to a constant PIHET efficiency of ~0.75%. We first modeled the transferred electron percentage as a function of pump fluence using a thermalized model assuming hot electrons are thermalized to a Fermi–Dirac distribution and only the portion with energy above RhB LUMO can transfer (Supplementary Note 6). The thermalized model yields a clear super-linear behavior (Fig. 4b), as has been generally observed in hot-electron thermionic emission devices based on metal or graphene[39]. The strong contrast between super-linear behavior based on thermalized electron distribution and experimentally observed linear relationship indicates PIHET occurs before electron thermalization process. This is consistent with much faster electron transfer rate than thermalization rate. The hot-electron distribution right after Landau damping is determined by pump photon energy and density of states[3,36]. The number of the hot electrons above RhB LUMO increases linearly with pump fluence at a certain photon energy (Fig. 4c). It is important to note we calculated the averaged photon number absorbed per NC <N> (Supplementary Note 7) and this value is much larger than 1 (Fig. 4a). Therefore, the linear power dependence is not due to linearly increased population of photoexcited NCs.

We also performed pump photon energy-dependent study. The obtained PIHET QY is estimated to be 1.4% for 0.855 eV or 1450-nm excitation. QY value decreases monotonically and almost linearly with reducing pump photon energy above a certain threshold (~0.68 eV) and then decreases slowly and approaches zero below that (Fig. 4b). This pump photon energy dependence is in striking contrast with direct transfer mechanism which should show a constant efficiency (Fig. 4b gray dashed line) but confirms indirect transfer mechanism where electron transfer occurs after Landau damping[22]. To confirm this and gain more physical insights, we modeled the pump photon energy-dependent QY by assuming only the portion ($\gamma$) of hot electrons above barrier height ($E_b$, the energy difference between FICO $E_f$ and RhB LUMO) after Landau damping can transfer to RhB with a transfer efficiency of $\eta$ (Fig. 4c and Supplementary Note 8), i.e., $QY = \eta\gamma$. The electron percentage ($\gamma$) above $E_b$ after Landau damping is determined by photon energy and $E_b$ (Fig. 4c). In principle, at zero temperature, no electrons from Landau damping can exceed $E_b$ if photon energy is less than that, which should set a distinct threshold value. Above that threshold, QY should increase linearly with photon energy and the slope of increasement is determined by transfer efficiency $\eta$. At room temperature (298 K), the threshold would be obscured by thermal distribution of electrons near fermi level and there would be a minor contribution extending below threshold. This is exactly

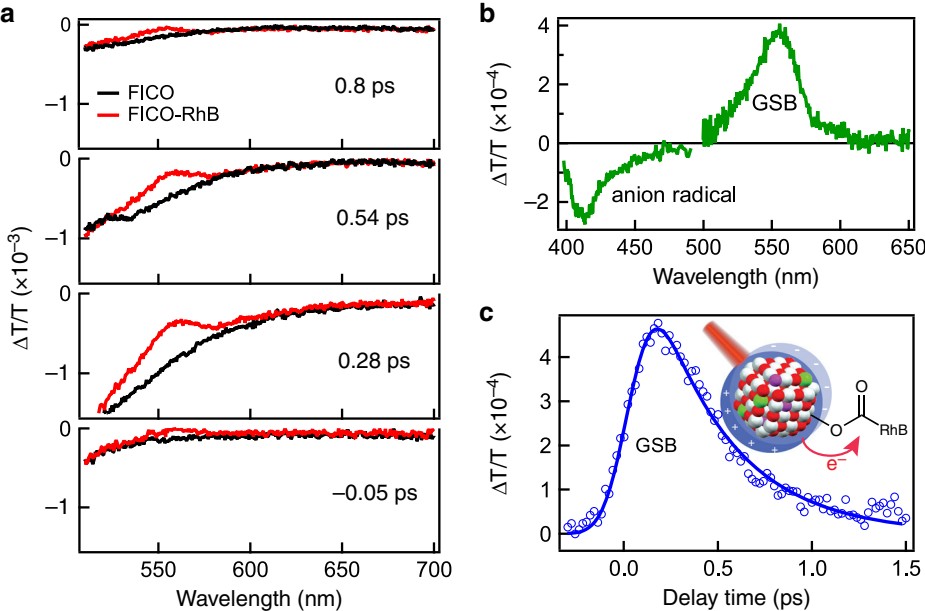

**Fig. 3 Hot-electron transfer from FICO. a** Comparison of transient absorption (TA) spectra at different delay times for FICO nanocrystals (NCs) and FICO–RhB. **b** Subtracted TA spectrum showing clearly RhB ground state bleach (GSB) and anion radical induced absorption (IA). **c** TA kinetics of RhB GSB kinetics (empty circles) and the exponential fitting (blue line) in FICO–RhB complex. Inset: scheme of PIHET from FICO NCs (red: O; white: Cd; green: In; violet: F) to RhB molecules. After infrared light excitation (red beam), the hot-electron transfers to the adsorbed RhB (red arrow).

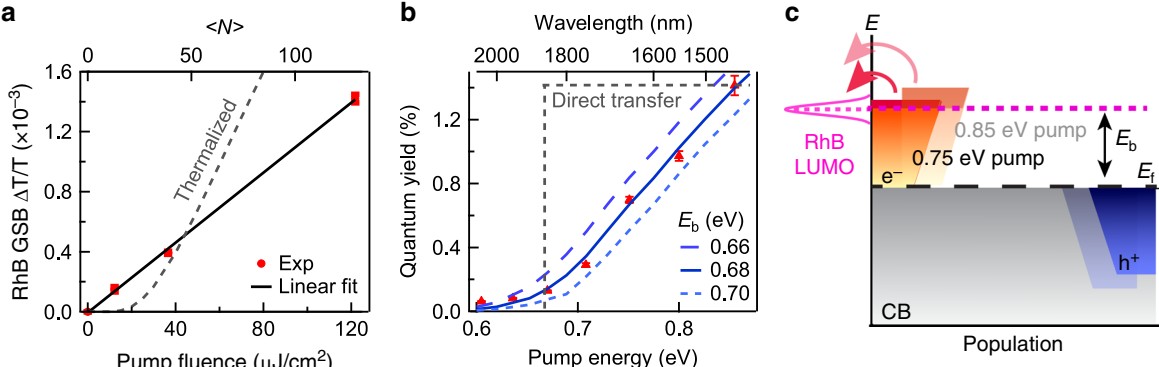

**Fig. 4 Mechanistic study of hot-electron transfer. a** RhB ground state bleach (GSB) signal (red symbols) as a function of pump fluence or averaged photon number per NC (<$N$>) and the linear fit (black line). Also shown in gray dashed line is modeled hot-electron transfer behavior with thermalized electron distribution, showing superliner behavior. **b** PIHET quantum yield (QY) as a function of pump photon energy. The red triangles are experimental results. The blue dashed line is modeled results with different barrier height $E_b$ and transfer efficiency $\eta$ of 5.5%. The gray dashed line shows the direct transfer behavior, contradicting with experimental results. **c** Scheme of hot-electron transfer with two pump energies (0.75 and 0.85 eV) after Landau damping before electron thermalization. The gray region shows the undisturbed filled states in conduction band. The orange and blue regions represent the excited electrons and empty states left, respectively, after Landau damping under different pump photon energies. Higher pump energy leads to more electrons above barrier height.

what we observed experimentally. In Fig. 4b, we show the modeled QY at 298 K with different $E_b$ values and $E_b$ of 0.68 eV provides the best agreement with experiments results. The extracted 0.68 eV barrier height is exactly same as the energy difference between estimated $E_f$ (−4.62 eV) and RhB LUMO (−3.94 eV), confirming the proposed picture. The experimental results can also be described with simple Fowler fit, but our model gives better mechanistic picture. (Supplementary Fig. 6 and Supplementary Note 8). From the modeling, we also obtained the transfer efficiency $\eta$ of 5.5%. The agreement between the modeled and experimental results presents a convincing picture of indirect hot-electron transfer after Landau damping but before electron thermalization. As shown in Fig. 4d, there is a distribution of excited hot electrons after Landau damping and higher pump

photon energy generates distribution extending to higher energy thus higher percentage above the barrier. Photoexciting interband transition of FICO in UV region has been performed before, which is not the focus in this study[40]. We also tuned excitation wavelength to 800 nm which is not in resonance with FICO or RhB transitions and observed no TA signal, precluding coherent artifacts from temporally overlapped pump and probe beams in sample. It would be desirable to perform photodegradation experiment on RhB as a proof-of-concept experiment. Unfortunately, we tried but had difficulty finding suitable electron sacrificial donor that can extract hole efficiently (<1 ps) from FICO without perturbing the surface attached RhB molecules. Based on the fast (<1 ps) back electron transfer from reduced RhB anion radical to FICO (Fig. 2c), the photodegradation efficiency

for this system is likely very small. Therefore, we focus on the transient reduction here for the mechanism studies.

## Discussion

Based on the mechanistic picture, the low PIHET efficiency from doped semiconductor NCs (~1.4% at 1450-nm or 0.85-eV pump, Fig. 4b) has two main origins: (i) only a certain percentage ($\gamma$) of hot electrons have energy above transfer barrier after Landau damping, and (ii) among that percentage of electrons, only a small portion ($\eta$) can transfer to electron acceptors. The former could be improved by raising fermi level of plasmonic NCs or lowering LUMO level of electron acceptors. For FICO–RhB complex, under 1450 nm pump, we estimated the percentage ($\gamma$) of electrons above RhB LUMO to be 25.9% and $\eta$ of 5.5% (Supplementary Note 7). We ascribe the second loss mainly to the steady state electron distribution in doped NC. Recently, Milliron et al. has shown a surface depletion layer in n-doped semiconductor NCs with up-bended band potential near the surface[41]. This layer will impede a large portion of hot electron generated inside NCs to reach surface, and the transferred portion of hot electrons are those generated on/near the surface and electronically coupled to electron acceptors. Fortunately, electron distribution in plasmonic NCs can be engineered through controlling the dopant distribution[42,43], and for example, can be concentrated near the surface of NCs, which should facilitate electron transfer process[44].

In summary, we have unambiguously demonstrated and thoroughly investigated PIHET from plasmonic metal oxide NCs with infrared response to electron accepting molecules using TA spectroscopy. We establish that the IA feature on TA spectra provides rich information on hot-electron generation, distribution and relaxation through e–e and e–ph scattering. After coupling to electron acceptors, hot electrons in plasmonic metal oxide NCs can transfer to accepting molecules in ultrafast timescale (<50 fs) right after Landau damping before electron thermalization, prevailing over energy loss from e–e and e–ph scattering. We unravel two main efficiency-limiting factors in semiconductor plasmonic nanocrystals. Our results pave the way for efficient hot-electron harvesting from non-noble metal plasmonic materials for infrared driven optoelectronics and photocatalysis.

## Methods

**Nanocrystal synthesis**. Chemicals: Cadmium (II) acetylacetonate (Cd(acac)$_2$, ≥99.9%), indium (III) fluoride (InF$_3$, ≥99.9%), oleic acid (OLAC, 90%), and anhydrous hexane (≥99%) were purchased from Sigma-Aldrich. Rhodamine B (RhB) and 1-octadecene (ODE, 90%) were purchased from Alfa Aesar. FICO NCs was synthesized following Ye's method with minor changes[26]. In a typical synthesis, 0.15 mmol of the total metal precursor with 20% of InF$_3$ and rest of Cd(acac)$_2$ and 0.48 mmol of OLAC were mixed with 5 ml ODE. The reaction solution was degassed under vacuum at 135 °C for 10 min and heated to ~316 °C. After 15–20 min refluxing, the solution turned to dark cyan. At 30 min, the solution was precipitated with isopropanol and washed with ethanol for twice. The NCs finally were dispersed in anhydrous hexane. To prepare FICO–RhB complex, about 50 mg of RhB powder was mixed with FICO solution and sonicated for 15 min. The solution turned from light cyan to pink. The mixture was filtrated to get a clean solution, as RhB is insoluble in hexane.

**Transient absorption measurement**. For femtosecond TA measurement, the fundamental beam (1030 nm) from Yb:KGW laser (20 W, 100 K Hz, Pharos, Light Conversion Ltd) was separated into two paths. One was introduced to an optical parametric amplifier to generate pump pulse at a certain wavelength in NIR (Light Conversion, OPA Orpheus-One). The other one was focused onto a YAG crystal to generate white light continuum from 500 to 900 nm or focused on a BBO crystal and then to a sapphire window to generate white light from 390 to 490 nm as probe light. The pump and probe beam overlapped inside the 1 mm cuvette at 30°. The Pump beam was sent to a motorized delay stage (Newport) and chopper at 25 kHz (Sci-Tec). The probe beam was collected by a CMOS detector (e2V) coupled to a spectrograph (Zolix). The instrument response function (IRF) was determined from solvent response to be 200 fs (Supplementary Fig. 7), which indicates the time

resolution of ~200/4 = 50 fs through IRF-convolution fitting. Pump and probe beam sizes were estimated using pinhole as 220 and 63.2 μm, respectively.

## Data availability

The source data necessary to support the findings of this paper are available from the corresponding author upon request.

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

## Acknowledgements

We thank the financial support from the National Natural Science Foundation of China (21773208, 21803055), the Fundamental Research Funds for the Central Universities and National Key Research and Development Program of China (2017YFA0207700).

## Author contributions

D.Z. and H.Z. designed the research; D.Z. performed the sample preparation and transient absorption experiments with the help of X.L. and Q.Z.; D.Z. and H.Z. analyzed data and wrote of the paper.

## Competing interests

The authors declare no competing interests.
