## [Peer Review File · Nature Communications]

Reviewers' comments:

Reviewer #1 (Remarks to the Author):

This is very interesting work on electron transfer dynamics from non-noble metal plasmonic nanocrystals to adsorbed molecules (rhodamine B), showing the detailed reaction rates (both for forward and back transfer) and the quantum yield (1.4%). The mechanism is well explored by careful experiments changing the excitation intensity and wavelength. The proposed reaction model seems reasonable and will be very much interesting and informative to many readers of this journal since plasmon induced electron transfer process is attractive in the areas of physics, chemistry, and material science and technology of nanoparticles. Also utilization of near-infrared light in energy conversion application is important as one of key technologies for solving global warming problems. Basically I would like to recommend this work for publication, but there are several requests as revision as listed below.

1. Figures 2a, 3a, 3b, 4a: The vertical axis does not indicate the location of absorbance = 0. I felt very difficult to understand the data. These graphs must be improved.

2. Page 6, line 106: "IA onset at ~ 2.74 eV or 453 nm" is not experimentally observed. I was confused with this discussion. Some correction or careful description is needed.

3. Figure 3a: Electron transfer reaction should produce an anion radical of RhB (RhB⁻). Some papers report that an absorption peak appears from 415 to 440 nm. I wonder why transient absorption spectral data does not show this spectral range. It was out of the measurable range? Authors should mention in the text why RhB⁻ could not be observed.

3. Figure 4a: TA kinetics of FICONCs at 550 nm and RhB bleach seem similar each other after ≈ 0.5 ps. Their lifetimes should be shown, and the mechanism should be discussed if they are identical. In general, charge recombination time does not relate to plasmon relaxation.

Reviewer #2 (Remarks to the Author):

In this manuscript, Zhou et al. explore infrared plasmon driven hot electron transfer from non-noble metal nanoparticles to surrounding attached RhB molecules. Understanding how hot carriers are transferred at chemical interfaces is a topic of growing interest and I think this work has some nice experiments that fall into this category. However, in my opinion, the manuscript in its current form is not rigorous enough to be published in Nature Communications. The following major points need to be addressed.

1. I wonder about the importance of this work. As I mentioned, hot electron transfer is an important topic and as such, there are many papers demonstrating and even quantifying hot electron transfer at various interfaces. Even with IR plasmons, a publication by Lian et al. (Nat. Commun. 9, 2314, 2018) already showed how hot holes transfer from CuS to CdS nanocrystals. See also a recent paper by Foerster et al. (Sci. Adv., DOI: 10.1126/sciadv.aav0704, 2019) that showed how molecules impact the plasmon damping and hot carrier transfer. As of now the authors have failed to justify how this work advances our knowledge of hot electron transfer as compared to these previous works. Is there any new physics here? Or is this electron transfer somehow fundamentally different? I don't see this as of now. For a journal like Nat. Commun., showing evidence for hot electron transfer for a new system is not sufficient.

2. Control experiments are missing. What happens when you excite FICO nanoparticles far from plasmonic resonance?

3. The evidence for hot electron transfer at FICO/RhB interfaces should be further supported through degradation experiments. RhB is a dye and I expect it should degrade if hot electrons are transferred to it from FICO.

4. The above experiment should also reveal if the degradation efficiency follows the absorption spectrum, i.e. degradation should be maximum at the plasmon resonance peak, around 1750nm.

5. I expect RhB to bind strongly with FICO through its carboxylic group as mentioned by the authors. How does this affect the LUMO? Strong hybridization can indeed change the LUMO/HOMO levels.

6. In Fig. 3c, the schematic is a bit misleading. After thermalization, the hot-electron tail extends to -3 eV, which is not possible energetically. The maximum energy above the Fermi level should be equal to the energy of the photon. Similarly, the hot-hole tail as well.

7. The authors state that in Fig. 3a, after 0.48 ps, both FICO and FICO/RhB show the same IA. But even at 0.74 ps, they are not the same.

8. Given that the rise time of the RhB bleach process is less than 50 fs, I speculate that direct transfer is indeed happening. Indirect transfer occurs usually at longer timescales. Low efficiency could mean that there is a large amount of back transfer taking place, but it doesn't necessarily point to indirect transfer. This point should be further explored.

Reviewer #3 (Remarks to the Author):

In this work, the authors use fluorine and indium co-doped CdO (FICO) nanoparticles to attempt to drive charge transfer between the nanoparticle's conduction band and an organic semiconductor, Rhodamine B (RhB), via plasmon induced hot electron transfer (PIHET). Photoexciting the nanoparticles' LSPR redistributes energy among their electrons, creating a population of high-energy electrons that can in principle transfer to RhB's LUMO level. Evidence of this exchange is shown experimentally using transient absorption spectroscopy, which suggests the timescale of electron transfer is faster than the electron cooling rate, < 50 fs. The authors calculate the efficiency FICO-to-RhB electron transfer as a function of both excitation fluence and pump photon energy and conclude both measurements suggest the transfer of hot carriers prior to electron thermalization.

Using metal oxide plasmonic nanocrystals to drive electron transfer is promising for a range of applications and certainly warrants attention. However, I find myself unconvinced that the authors observe PIHET as there are several effects that could lead to photobleaching of RhB upon FICO excitation, chief of which is a change in RhB's absorption spectrum due to the Stark Effect. Extending the authors' probe source to see features indicative of formation of RhB anions is critical to prove successful charge transfer to RhB. In addition, fitting the Fowler equation to the author's data over a wider energy variation range that includes excitation pulses below the expected barrier for electron transfer is also needed to show that RhB bleaching does not arise simply from a Stark effect. Without such data, I am dubious of the authors' claims of PIHET.

Below, I have included some additional points for the authors to consider when revising their manuscript.

1. The biggest question for this report is the “unambiguous” evidence of electron transfer seen in figure 3a. Seeing a bleach of the RhB feature could denote electron transfer, but it could also stem from various physical changes occurring around the RhB molecule. The RhB shifting could be due to the Stark effect, where the LSPR excitation heats charge carriers and these electrons change the dielectric environment of the RhB molecule, causing its max absorption peak to shift. A much cleaner assignment of PIHET would come from observation of induced absorption features from the RhB anion. The extinction of this species could be determined under steady-state conditions using spectroelectrochemistry experiments. Comparing such experiments against transient absorption data would also provide a cleaner way of assessing the PIHET yield, if indeed this process underlies the authors’ results.

2. It would be useful to see equations and graphs that describe the calculated DOS and the changing Fermi-Dirac distribution used by the authors to compute the transient response of photoexcited FICO nanocrystals (section S5 of the supporting information). Is this model able to reproduce the UV-vis spectrum of the particles shown in figure 1c? I ask in part as the Tauc plot in this figure implies there is significant absorption below the NC Fermi level, which I suspect may not be able to be fully described by a room temperature Fermi-Dirac distribution.

3. It would be good to quantify if the HOMO and LUMO levels of RhB shift when they are adsorbed to the surface of FICO as there is evidence of this happening with other molecules. How confident are the authors that the literature values for the placement of these bands in solution reflect their values when adhered to FICO nanoparticles? Presumably, extending the Fowler plot in figure 4c would allow the authors to better assess the LUMO level position of RhB molecules when adhered to FICO.

4. How much information do the authors have regarding the amount of RhB that binds to each nanoparticle? How are RhB molecules adhered to the FICO surface and is there expected to be a preferred geometry with which they bind? I ask as it would be helpful to know the distance scale over which electron transfer from FICO to RhB needs to take place and the density of acceptor molecules held at that distance to make electron transfer competitive with electron cooling.

5. Some additional details regarding the transient absorption layout used by the authors is warranted. The authors report use of a Pharos system, but what is the repetition rate used for transient absorption experiments? Likewise, what detector was used for reading out data? The authors also quote their instrument response function as being 290 fs, yet highlight time dynamics and spectral shifting in the main text at time delays shorter than this in figures 2, 3, and 4, such as a 50 fs rise of RhB bleaching. What is the reason the authors feel confident that such dynamics are simply not an experimental artifact tied to other nonlinear processes that can occur when the pump and probe pulses are temporally overlapped in the sample?

6. The authors note a delayed rise of the data at 550 nm that they attribute to electron thermalization within the FICO conduction band. Showing data that fully highlights the development of this thermalized distribution (such as a contour map of the full spectral dynamics of the TA data) would be useful as well as a fit to the data from the authors' spectral model for electron thermalization.

7. In figure 4c, is the efficiency plotted an internal quantum efficiency that accounts for the FICO nanoparticle absorption strength at each probe energy or is it an external quantum efficiency that does not? I am not sure if the Fowler equation considers spectral variations in absorption strength, which could explain the differences in curvature with changing photon energy between the Fowler equation prediction and that shown by the experimental data. Also, as I mentioned in point 3, it would be good to extend this plot over a wider photon excitation energy range to demonstrably show that electron transfer fully turns off once the photon energy is reduced below the threshold for electron transfer set by the energy difference between the FICO Fermi energy and RhB LUMO level.

8. What is the difference between the underlying assumptions made by the Fowler model and those used to produce the blue trace in figure 4c? I think the assumptions are very similar, so I'm not sure what the benefit is for showing both models.

9. In section S3 of the supporting information, why is $T_{e,i}$ referenced as the "ideal" electron temperature? Why is it "ideal"? Is there a difference between your calculated $T_{e,i}$ and the $T_{e,i}$ you get by fitting the TA data?

We thank all reviewers for the constructive feedback and stimulating comments to improve the manuscript. We have performed all new experiments requested and addressed questions and concerns raised by reviewers in the revised manuscript. The manuscript has been improved significantly. Following the questions and comments, we made the following corrections and improvements point by point in response. **The responses are in red and revisions are in blue.** The manuscript is revised using track-change mode and the marked version is submitted for review. The page number is referred to marked version.

Reviewer #1 (Remarks to the Author):

This is very interesting work on electron transfer dynamics from non-noble metal plasmonic nanocrystals to adsorbed molecules (rhodamine B), showing the detailed reaction rates (both for forward and back transfer) and the quantum yield (1.4%). The mechanism is well explored by careful experiments changing the excitation intensity and wavelength. The proposed reaction model seems reasonable and will be very much interesting and informative to many readers of this journal since plasmon induced electron transfer process is attractive in the areas of physics, chemistry, and material science and technology of nanoparticles. Also utilization of near-infrared light in energy conversion application is important as one of key technologies for solving global warming problems. Basically I would like to recommend this work for publication, but there are several requests as revision as listed below.

Response: We appreciate reviewer for the positive comments and sharing the same enthusiasm on the importance and broad impact of this study as us.

1. Figures 2a, 3a, 3b, 4a: The vertical axis does not indicate the location of absorbance = 0. I felt very difficult to understand the data. These graphs must be improved.

Response: we are sorry for the confusion. We have re-made all figures in the revised manuscript with Y axis values labeled clearly and spectral evolution shown clearly.

2. Page 6, line 106: "IA onset at ~ 2.74 eV or 453 nm" is not experimentally observed. I was confused with this discussion. Some correction or careful description is needed.

Response: We are sorry for the confusion in the previous submission.

1) The 2.74 eV or 453nm IA onset described in the original manuscript is the estimated theoretical value based on energy diagram and photon energy. Because of electron doping, the interband optical bandgap is 3.49 eV determined from absorption spectrum. Considering the 0.75 eV photon energy, Landau damping should create an interband IA starting from $3.49 - 0.75 = 2.74$ eV.

2) in the revised manuscript, we have performed and added new TA results with probe extending to 400 nm. Now we can have a full view on the electron thermalization process. Experimentally, the initial IA indeed starts at ~ 450 nm (see 0.14 ps spectra in Fig. 2a) and then progressively shifts to 670 nm, due to electron thermalization process. The experimental IA onset agrees very well with estimated value from Landau damping, validating the picture.

Revisions: 1) we changed Fig. 2a in the original manuscript to the following one which shows initial IA onset at ~ 450 nm after Landau damping.

2) in page 6, second paragraph, we revised as:

Considering the conservation of photon energy (1650 nm or 0.75 eV)¹ and the relaxed momentum conservation for NCs, the lowest energy electron that can be excited is 0.75 eV below E_f , which should generate an interband IA starting from ~ 2.74 eV ($= 3.49 - 0.75$) or 453 nm (Fig. 2c left panel). This estimated value agrees very well with experimentally observed ~ 450 nm IA onset at 0.14 ps, confirming the proposed picture (Fig. 2C left panel).

3. Figure 3a: Electron transfer reaction should produce an anion radical of RhB (RhB⁻). Some papers report that an absorption peak appears from 415 to 440 nm. I wonder why transient absorption spectral data does not show this spectral range. It was out of the measurable range? Authors should mention in the text why RhB⁻ could not be observed.

Response: We thank the reviewer raising this key missing point in the previous submission. At that time, we only generated white continuum with 1030 nm fundamental light which can only reach 500 nm at blue side therefore we missed a lot of information. We recently added the continuum generation using 515 nm (SHG of 1030 nm) and now we can access the wavelength range between $390 - 480$ nm. We performed the TA experiment and observed the anion radical of RhB in $400-440$ nm^{2,3} and GSB of RhB simultaneously (see figure below), confirming electron transfer from photoexcited FICO to RhB.

Due to its weaker amplitude, absence of precise extinction coefficient and severe overlap with the strong induced absorption signal of FICO, in this study, we still use RhB GSB signal for quantitative analysis.

Revisions: 1) In SI, we add figure S5 showing extracting RhB anion radical signal from subtraction. In main content, Fig 3, we added full TA spectra from 400-650 showing both RhB GSB and anion radical simultaneously.

2) in main text, page 9 last paragraph, we revised

Subtracting 0.4 ps TA spectrum of FICO-RhB by that of FICO yields a difference spectrum in Fig. 3b. It shows a strong ground state bleach (GSB) peak matching exactly with RhB absorption) and an IA peak at ~ 412 nm which can be attributed to RhB anion radical (Supplementary Figure 5). We didn't observe any shifting or broadening signature of RhB absorption on TA spectra, indicating negligible contribution from dielectric environment change or Stark effect. This will be further confirmed with excitation wavelength dependent study later. Because 1650 nm cannot excite RhB molecule and hole transfer is energetically unfavored, the simultaneous formation of RhB GSB and anion radical IA indicate PIHET from photoexcited FICO to surface adsorbed RhB molecules (Fig. 3c inset). Because RhB anion radical is weaker without known extinction coefficient and overlaps with strong IA signal from FICO, we use RhB GSB for following quantitative analysis.

4. Figure 4a: TA kinetics of FICONCs at 550 nm and RhB bleach seem similar each other after ≈ 0.5 ps. Their lifetimes should be shown, and the mechanism should be discussed if they are identical. In general, charge recombination time does not relate to plasmon relaxation.

Response: We are sorry that the original plot in previous submission causes such confusion. As reviewer pointed out, actually there are very different physical processes, charge recombination and plasmon decay. They just all occur fast.

In the revised manuscript, we've re-made all figures, with plasmon decay in Fig. 2 and charge recombination in Fig. 3 where they should be. And we fit all these kinetics to extract the decay

process, the plasmon decays with a lifetime of 0.21 ps and charge recombination with a lifetime of 0.41 ps. Their lifetimes are different.

Revisions: 1) We have re-made Fig. 2b, Fig. 3c and Fig. 4 in main content.

2) we specify the plasmon relaxation lifetime and charge recombination lifetime from single exponential fitting.

page 7, we added:

A representative IA kinetics at 550 nm is shown Fig. 2b and the rising (decay) process corresponds to electron thermalization (cooling) process. We fit the kinetics with an exponential rise and decay function convoluted with IRF. It reaches maximum at ~ 0.5 ps with an electron thermalization lifetime of 150 fs and decays to zero in ~ 1.5 ps with cooling lifetime of 0.21 ps.

page 8, we added:

The GSB kinetics of RhB is shown in Fig. 3c and we fit it with an exponential rise and decay function convoluted with IRF. The lifetime of rise and decay process of RhB bleach kinetics are < 50 fs and 407 fs, respectively, corresponding to electron transfer to RhB and subsequent back electron transfer process.

Reviewer #2 (Remarks to the Author):

In this manuscript, Zhou et al. explore infrared plasmon driven hot electron transfer from non-noble metal nanoparticles to surrounding attached RhB molecules. Understanding how hot carriers are transferred at chemical interfaces is a topic of growing interest and I think this work has some nice experiments that fall into this category. However, in my opinion, the manuscript in its current form is not rigorous enough to be published in Nature Communications. The following major points need to be addressed.

Response: We appreciate reviewer for the positive comments about this interesting topic. We have performed substantial new experiments and analysis to improve our manuscript as shown below.

1. I wonder about the importance of this work. As I mentioned, hot electron transfer is an important topic and as such, there are many papers demonstrating and even quantifying hot electron transfer at various interfaces. Even with IR plasmons, a publication by Lian et al. (Nat. Common. 9, 2314, 2018) already showed how hot holes transfer from CuS to CdS nanocrystals. See also a recent paper by Foerster et al. (Sci. Adv., DOI: 10.1126/sciadv.aav0704, 2019) that showed how molecules impact the plasmon damping and hot carrier transfer. As of now the authors have failed to justify how this work advances our knowledge of hot electron transfer as compared to these previous works. Is there any new physics here? Or is this electron transfer

somehow fundamentally different? I don't see this as of now. For a journal like Nat. Commun., showing evidence for hot electron transfer for a new system is not sufficient.

Response: We thank reviewer raising question about the significance and pointing out the missing references. As we have discussed and cited in the revised manuscript, Teranishi's group has pioneered the plasmon induced charge transfer at semiconductor interface in various inorganic semiconductor nano-heterostructures.^{4, 5, 6} (including the Nat. Common. 9, 2314, 2018 reviewer pointed out) They have nicely synthesized those nano-heterostructures and demonstrated the plasmon induced charge transfer at inorganic hetero-interface and their applications in photochemical reactions. Here we focus on another generally existed interface and process in photochemical and photocatalytic system, that is hot electron transfer from plasmonic materials directly to molecules. This is direct plasmon driven hot carrier reduction/oxidation reaction. These are fundamentally different families of systems. In those nano-heterostructures, two semiconductors are grown together and strongly coupled with large donor and acceptor density of states. There, electron transfer occurs more like a transport process with possibility of direct transfer or ballistic transport or trapped mediated transport. On the other hand, in semiconductor-molecular interface, they are truly distinctly different donor-acceptor two-state system with weak electronic interaction/coupling and discrete molecular states. This imposes more challenges for two-state charge transfer. The plasmon charge transfer in this semiconductor-molecule interface behaves very differently at semiconductor-semiconductor interface.

In addition, not only reporting the direct plasmon hot electron reduction of molecules, in this study, by modelling the spectra carefully, we use this semiconductor-molecule model system show clear how the photon energy transform to nonthermalized hot carrier from initial Landau damping and then transform to thermalized hot electron Fermi-Dirac distribution through electron-electron scattering and finally cool through electron-phonon scattering. More importantly, by careful power and energy dependent studies and modeling, we show charge transfer at this semiconductor-molecule interface occurs after first stage (Landau damping) but before second state (electron thermalization). We draw the physical pictures, identified the efficiency limiting pathways and provide the designing strategies in generally existed plasmonic semiconductor-molecule donor-acceptor system.

As we focus on not only the hot electron transfer but also generation, we change our title slightly to "Infrared Driven Hot Electron Generation and Transfer from Non-Noble Metal Plasmonic Nanocrystals".

Another reference pointed out by reviewer focus on the plasmon optical properties (linewidth) perturbed by molecule dipoles. We didn't observe the plasmon peak change in our system.

Revisions:

We change our title slightly to “Infrared Driven Hot Electron Generation and Transfer from Non-Noble Metal Plasmonic Nanocrystals”

page 2, bottom, we revised,

Despite these unique properties, however, hot electron harvesting and conversion from doped semiconductor-based plasmon has been rarely reported, except a few recent studies of plasmon induced charge transfer in semiconductor nano-heterostructures and their applications in photocatalysis and photodetections. Plasmon driven charge injection from plasmonic semiconductors to molecules remains to be established.

page 3, middle, we revised,

we used F and In co-doped CdO (FICO) NCs as a model plasmonic semiconductor NC and Rhodamine B (RhB) as electron accepting molecules and investigated ultrafast generation and relaxation of hot electrons in plasmonic semiconductor NCs and hot electron injection to molecular acceptors.

2. Control experiments are missing. What happens when you excite FICO nanoparticles far from plasmonic resonance?

Response: We thank reviewer pointing out this missing control experiment. The absorption of FICO NC contains a NIR plasmon absorption and an interband absorption in UV region (as shown in Fig 1). We have performed the control experiment with two different excitation wavelengths.

We first chose the excitation wavelength at 360 nm at the inter-band absorption. The TA spectra of FICO NCs after 360 nm interband excitation shows an induced absorption different from plasmon excitation. Here, the induced absorption corresponds to the modulation of NIR plasmon resonance by ultrafast photodoping effect, which has been reported before.¹¹ For FICO-RhB under 360 nm excitation, we observed an induced absorption from FICO itself plus a bleached RhB signal which is likely due to the direct excitation of RhB since RhB also has absorption at 360 nm. The interband photodoping on FICO has been studied before.¹¹ In this study, we focus on NIR plasmon excitation and driven electron transfer.

We also choose an excitation wavelength (800nm) between NIR plasmon absorption and UV interband absorption and observed no TA signal, which precludes any coherent artifact

contribution from temporally overlapped pump and probe beams.

Revisions: main content, page 14, top, we added

Photoexciting interband transition of FICO in UV region has been performed before, which is not the focus in this study.¹¹ We also tuned excitation wavelength to 800 nm which is not in resonance with FICO or RhB transitions and observed no TA signal, precluding coherent artifacts from temporally overlapped pump and probe beams in sample.

3. The evidence for hot electron transfer at FICO/RhB interfaces should be further supported through degradation experiments. RhB is a dye and I expect it should degrade if hot electrons are transferred to it from FICO.

4. The above experiment should also reveal if the degradation efficiency follows the absorption spectrum, i.e. degradation should be maximum at the plasmon resonance peak, around 1750nm.

Response for 3 and 4: We thank reviewer for suggestion of photodegradation experiment. We have tried it very hard but have a few major problems preventing us obtain reliable and meaningful results.

1) unlike conventional photodegradation experiment, here, RhB molecules bind to FICO surface through the carboxylic group and RhB molecules cannot dissolve in nonpolar solvent (e.g. hexane) which we use for disperse FICO NCs. Therefore once electron inject to RhB, it will come back to FICO quickly in ~ 1 ps (as shown by the kinetics in figure below middle panel). This is charge recombination or back electron transfer process. Therefore, nothing changes in this photoexcitation cycle and we cannot reduce RhB accumulatively. There is no net effect.

2) To drive RhB photodegradation, we would need an electron sacrificial donor that can reduce the hole in FICO fast enough (< 1 ps). That is to say, the redox potential of electron sacrificial donor needs to be higher than fermi level (~ -4.6 eV) of FICO (Figure below left panel). And this molecule can dissolve in nonpolar solvent and approach FICO surface. We have searched/tried the electron sacrificial donor extensively and most of conventional electron sacrificial donor cannot meet these requirements. A potential one is alkyl thiol (e.g. $\text{CH}_3\text{-(CH}_2\text{)}_x\text{-SH}$) which has redox potential ~ -4.4 eV and can be solved in nonpolar solvent. But when we add alkyl thiol in solution, we notice the binding of alkyl thiol to FICO surface affects RhB binding and can replace the RhB molecules. Therefore, we see gradual decrease of RhB absorption with time. Because of fast charge recombination process, the efficiency of such photodegradation is expected to be very low. We cannot say whether and how much the decrease of RhB absorption (figure below right panel) is indeed due to photodegradation at current stage. It's unsafe to do quantitative experiment and analysis.

3) This study is more on the mechanisms, e.g. finding the physical pictures and the efficiency-limiting factors. Using transient spectroscopy study, we can transiently reduce RhB and perform all quantitative analysis and draw the physical picture without complication from electron sacrificial donor.

Revisions: in main content, page 14, middle, we added

It would be desirable to perform photodegradation experiment on RhB as a proof-of-concept experiment. Unfortunately, we tried but had difficulty finding suitable electron sacrificial donor that can extract hole efficiently ($< 1 \text{ ps}$) from FICO without perturbing the surface attached RhB molecules. Based on the fast ($< 1 \text{ ps}$) back electron transfer from reduced RhB anion radical to FICO (Fig. 2c), the photodegradation efficiency for this system is likely very small. Therefore, we focus on the transient reduction here for the mechanism studies.

5. I expect RhB to bind strongly with FICO through its carboxylic group as mentioned by the authors. How does this affect the LUMO? Strong hybridization can indeed change the LUMO/HOMO levels.

Response: We thank reviewer raising question about interaction between FICO and RhB. We expect no strong hybridization between FICO and RhB based on the absorption spectra. Strong hybridization will significantly damp the plasmon resonance and broaden the plasmon peak, as shown in previous study.¹² As shown in Fig. 1a or , the plasmon peak is changed by less than 10% after RhB binding. We also compared the absorption spectrum of RhB in FICO-RhB complex and in methanol solution. We redshifted the wavelength of RhB solution by 7 nm because of different solvent polarity. The RhB peak width in FICO-RhB and RhB solution is also quite similar. These results suggest no strong hybridization between FICO and RhB. The absence of strong hybridization is also supported by indirect transfer mechanism instead of direct transfer.

2) following 3rd reviewer's suggestions, we have extended our excitation energy to even lower energy (0.605 eV or 1950 nm, 0.636 eV or 2050 nm). The quantum yield (QY) as a function of excitation energy are shown in Fig. 4b. The QY decreases monotonically with decreasing energy and shows a knee-behavior at ~ 0.68 eV. Below 0.67 eV, there is still measurable QY but the value is very small, which is likely due to sample heterogeneity. The results above 0.68 eV can be very well modeled assuming only the percentage of electrons with energy above barrier height after Landau damping can transfer to RhB LUMO (Fig. 4c). Based on the modeling, the barrier height is ~ 0.68 eV. The theoretical barrier height should be -3.94 (LUMO of RhB) - (-4.62) (Ef of FICO) = 0.68 eV. The E_b determined from pump energy dependent study is same as the theoretical value, which confirms the position of RhB LUMO level as in literatures.

Revisions: in main content, page 4, top, we added

The absence of significant broadening of FICO plasmon resonance and RhB absorption (Supplementary Figure 2) precludes the strong hybridization between FICO and RhB.

in main content, we have revised Fig. 4b with results from lower photon energies added. page 13 bottom, we revised

The electron percentage (γ) above E_b after Landau damping is determined by photon energy and E_b (Fig. 4c). In principle, no electrons from Landau damping can exceed E_b if photon energy is less than that, which sets a threshold value. Above that threshold, QY should increase with photon energy and the slope of increasement is determined by transfer efficiency η . In Figure 4b, we show the modeled QY with different E_b values and E_b of 0.68 eV provides the best agreement with experiments results. The extracted 0.68 eV barrier height is same as the energy difference between estimated E_f (-4.62 eV) and RhB LUMO (-3.94 eV), confirming the proposed picture.

6. In Fig. 3c, the schematic is a bit misleading. After thermalization, the hot-electron tail extends to -3 eV, which is not possible energetically. The maximum energy above the Fermi level should be equal to the energy of the photon. Similarly, the hot-hole tail as well.

Response: We are sorry we did not present it clearly in the previous manuscript. The review's statement that "The maximum energy above the Fermi level should be equal to the energy of the photon" is correct, just as shown by the left panel in Fig. 3c. This is the situation right after

Landau damping in 10 fs where the maximum energy of the hot electron is equal to the energy of the photon, i.e. 0.75 eV about the Fermi level. Same thing is for hot-hole, 0.75 below the Fermi level. Indeed, we experimentally observed the IA onset at ~ 450 nm right after photoexcitation, which confirms the scheme in Fig. 3c left panel.

After initial non-thermalized hot electrons formed from Landau damping in 10 fs, electron thermalization process starts through e-e and e-ph scattering. Nonthermalized hot electrons reaches thermalization with Fermi-Dirac distribution in ~ 0.5 ps with a characteristic temperature. The electrons exchange energies without limitation from photon energy anymore. The profile of electron distribution is determined solely by this temperature and can reach a high energy for thermalized hot electrons (Fig. 2c right panel).

The temperature is determined by the pump power. In our experiment in Fig. 2a, assuming no energy loss to phonon, the theoretical maximum electron temperature is estimated to be 5763 K. The temperature extracted from experimental IA of thermalized electron distribution is 4500 K. (see details in main content). At these temperatures and room temperature (300 K), the carrier distribution is as below, which shows electron can reach high energy (much higher than photon energy) at high temperature, much higher than initial photon energy.

Revisions: 1) in Fig. 2c life panel, we modified the figure to label photon energy (E_{photon}) after Landau damping. For the thermalize distribution, it can reach higher energy (Fig. 2c right panel).

2) In Fig. 2a, we added TA spectra from 400-490 showing full IA spectrum from early time after Landau damping to electron thermalization. The initial IA spectrum shows an onset at ~ 450 nm, corresponding to electron distribution after Landau damping. The IA onset reaches 675nm, corresponding to thermalized electron distribution. These values agree with the proposed pictures quantitatively.

7. The authors state that in Fig. 3a, after 0.48 ps, both FICO and FICO/RhB show the same IA. But even at 0.74 ps, they are not the same.

Response: We apologize for not describing it clearly in the previous manuscript. TA of FICO/RhB contains IA from FICO and GSB from RhB. They are only same at longer wavelength (> 600 nm) where RhB doesn't absorb. At shorter wavelength (< 600 nm), the difference between the IA signal for FICO and FICO-RhB is the RhB GSB signal. We revised our statement in manuscript to make it clear.

Revisions: page 8, bottom, we revised as

“After photoexcitation under same experimental conditions, FICO and FICO-RhB show almost same IA spectra at wavelength range (> 600 nm) where RhB doesn't absorb (Fig. 3a), suggesting little energy dissipated thus low electron transfer efficiency.”

8. Given that the rise time of the RhB bleach process is less than 50 fs, I speculate that direct transfer is indeed happening. Indirect transfer occurs usually at longer timescales. Low efficiency could mean that there is a large amount of back transfer taking place, but it doesn't necessarily point to indirect transfer. This point should be further explored.

Response: We thank reviewer pointing this interesting and important idea. Yes, the electron transfer lifetime (< 50 fs) is beyond our time resolution limit. Similar ultrafast electron transfer has been reported before in dye sensitized semiconductor oxides, which is actually not associated with direct transfer mechanism.¹³ We think the indirect transfer mechanism is the situation here based on the following two reasons:

1) the direct transfer occurs through the plasmon dephasing process through Landau damping which is at a few fs. This should lead to significantly (a few times) broadening of plasmon resonance as shown in previous study.¹² But in our sample, FICO-RhB only shows a slightly broadening (< 10%), which means RhB doesn't affect the plasmon dephasing process. There is negligible direct transfer.

2) The more direct evidence comes from the pump energy dependent study. For direct transfer, the pump energy dependent efficiency would be constant, without any dependence on pump energy.¹² But In our sample, the pump energy dependent efficiency increases at higher energy after a certain threshold. This means charge transfer occurs after plasmon dephasing process, that is, indirect transfer picture.

Revisions: 1) Main content, figure 4b, we added the pump energy dependent QY behavior, contradicting with experiment result.

2) main content, page 10, bottom

Direct transfer through plasmon dephasing process usually manifests as high efficiency and significantly broadened plasmon resonance.¹² The low efficiency, together with only slightly (< 10%) broadened plasmon peak observed in FICO-RhB complex, suggests PIHET in FICO-RhB

likely occurs through indirect mechanism with hot electron generation from Landau damping followed by transfer process. This will be further investigated and confirmed below.

3) main content, page 12, bottom

This dependence is in striking contrast with direct transfer mechanism which should show a constant efficiency (Fig. 4b gray dashed line) but confirms indirect transfer mechanism where electron transfer occurs after Landau damping.

Reviewer #3 (Remarks to the Author):

In this work, the authors use fluorine and indium co-doped CdO (FICO) nanoparticles to attempt to drive charge transfer between the nanoparticle's conduction band and an organic semiconductor, Rhodamine B (RhB), via plasmon induced hot electron transfer (PIHET). Photoexciting the nanoparticles' LSPR redistributes energy among their electrons, creating a population of high-energy electrons that can in principle transfer to RhB's LUMO level. Evidence of this exchange is shown experimentally using transient absorption spectroscopy, which suggests the timescale of electron transfer is faster than the electron cooling rate, < 50 fs. The authors calculate the efficiency FICO-to-RhB electron transfer as a function of both excitation fluence and pump photon energy and conclude both measurements suggest the transfer of hot carriers prior to electron thermalization.

Using metal oxide plasmonic nanocrystals to drive electron transfer is promising for a range of applications and certainly warrants attention. However, I find myself unconvinced that the authors observe PIHET as there are several effects that could lead to photobleaching of RhB upon FICO excitation, chief of which is a change in RhB's absorption spectrum due to the Stark Effect. Extending the authors' probe source to see features indicative of formation of RhB anions is critical to prove successful charge transfer to RhB. In addition, fitting the Fowler equation to the author's data over a wider energy variation range that includes excitation pulses below the expected barrier for electron transfer is also needed to show that RhB bleaching does not arise simply from a Stark effect. Without such data, I am dubious of the authors' claims of PIHET.

Response: We sincerely thank reviewer's agreement on the significance and impact of this study and the critical suggestions on more experimental evidence. We have performed all the experiments following reviewer's suggestion. In the revised manuscript, we have added the wavelength probe region between 400-480 where we clearly show the formation of RhB anion radical in FICO-RhB and the shifting of IA onset to show the electron thermalization process. We have also added more low pump photon energies (0.605 eV or 1950 nm, 0.636 eV or 2050 nm) in the pump energy dependence study. All these results indicate hot electron transfer to RhB, instead of other effect, e.g. stark effect. Below are the details.

Below, I have included some additional points for the authors to consider when revising their manuscript.

1. The biggest question for this report is the “unambiguous” evidence of electron transfer seen in figure 3a. Seeing a bleach of the RhB feature could denote electron transfer, but it could also stem from various physical changes occurring around the RhB molecule. The RhB shifting could be due to the Stark effect, where the LSPR excitation heats charge carriers and these electrons change the dielectric environment of the RhB molecule, causing its max absorption peak to shift. A much cleaner assignment of PIHET would come from observation of induced absorption features from the RhB anion. The extinction of this species could be determined under steady-state conditions using spectroelectrochemistry experiments. Comparing such experiments against transient absorption data would also provide a cleaner way of assessing the PIHET yield, if indeed this process underlies the authors’ results.

Response: We thank the reviewer raising this key missing evidence in the previous manuscript. At that time, we only generated white continuum with 1030 nm fundamental light which can only reach 500 nm at blue side therefore we missed a lot of information. We recently added the continuum generation using 515 nm (SHG of 1030 nm) and now we can access the wavelength range between 390 - 480 nm. We performed the TA experiment and observed the anion radical of RhB in 400-440 nm^{2,3} and GSB of RhB simultaneously (see Figure below), confirming electron transfer from photoexcited FICO to RhB.

The effect of dielectric environment change or dipole/electric field effect from photoexcitation should lead to the shifting and broadening of RhB ground state absorption in TA measurement. Considering the molecular dipole distribution, the ground state absorption of RhB should show a combined peak broadening and redshifting. We didn’t see any signature of broadening or redshifting of RhB absorption in our TA spectra. Instead, we simply observe a ground state bleach, which indicates negligible other effects. We also performed TA measurement with even lower photon energy (0.605 eV or 1950 nm, 0.636 eV or 2050 nm) where FICO still absorbs strongly but electron transfer is energetically unfavored (Fig. 4b). We observe very little RhB GSB signal, which also precludes dielectric environment change or dipole/electric field effect from photoexcitation. We also performed TA measurement with 800 nm where both FICO and RhB don’t absorb and observed no TA signal, precluding coherent artifacts where pump and probe beam temporarily overlap.

Having established electron transfer reaction, due to anion radical’s weaker amplitude, absence of precise extinction coefficient and severe overlap with the strong induced absorption signal of FICO, in this study, we still use RhB GSB signal for quantitative analysis.

Revisions: 1) In SI, we add figure S5 showing extracting RhB anion radical signal from subtraction. In main content, Fig 3, we added full TA spectra from 400-650 showing both RhB GSB and anion radical simultaneously.

2) in main text, page 9 last paragraph, we revised

Subtracting 0.4 ps TA spectrum of FICO-RhB by that of FICO yields a difference spectrum in Fig. 3b. It shows a strong ground state bleach (GSB) peak matching exactly with RhB absorption) and an IA peak at ~ 412 nm which can be attributed to RhB anion radical (Supplementary Figure 5). We didn't observe any shifting or broadening signature of RhB absorption on TA spectra, indicating negligible contribution from dielectric environment change or Stark effect. This will be further confirmed with excitation wavelength dependent study later. Because 1650 nm cannot excite RhB molecule and hole transfer is energetically unfavored, the simultaneous formation of RhB GSB and anion radical IA indicate PIHET from photoexcited FICO to surface adsorbed RhB molecules (Fig. 3c inset). Because RhB anion radical is weaker without known extinction coefficient and overlaps with strong IA signal from FICO, we use RhB GSB for following quantitative analysis.

2. It would be useful to see equations and graphs that describe the calculated DOS and the changing Fermi-Dirac distribution used by the authors to compute the transient response of photoexcited FICO nanocrystals (section S5 of the supporting information). Is this model able to reproduce the UV-vis spectrum of the particles shown in figure 1c? I ask in part as the Tauc plot in this figure implies there is significant absorption below the NC Fermi level, which I suspect may not be able to be fully described by a room temperature Fermi-Dirac distribution.

Response: We are sorry we didn't make it clear in the previous submission.

1) The Fermi-Dirac distribution is given by

$$f(E) = \frac{1}{1 + e^{(E-E_f)/kT_e}}$$

where E_f is the estimated fermi level (-4.62 eV) from optical absorption, T_e is the electron temperature. Assuming parabolic conduction band dispersion and flat valance band, the relative induced absorption could be calculated by the change of Fermi-Dirac distribution (pumped vs unpumped) multiplying the conduction band dispersion

$$\frac{\Delta T}{T}(E) \propto (f(E) - 1) * \sqrt{E + 6.25}$$

-6.25 is where CB starts from. We use above equation to simulate the transient spectra after electron thermalization with a characteristic temperature T_e .

2) The absorption tail extends to ~ 0.3 eV below Tauc plot cutoff. This is more than the thermal distribution broadening (~ 0.1 eV) (see below). The absorption tail likely contains contribution from sample heterogeneity, especially different doping levels or fermi levels.

Revisions:

1) we added SI3 section to describe details of calculating IA spectra with all equations provided.

2) in main content, page 5, middle, we added

There is an absorption tail extending to ~ 0.3 eV below $B_{g,opt}$, which is likely due to combined contribution from thermal broadening and sample heterogeneity.

page 8, top, we revised

We modeled the IA spectra after electron thermalization (0.5 ps) to extract the electron temperature (T_e) based on Fermi-Dirac distribution and parabolic band dispersion (Supplementary Note 3).

3. It would be good to quantify if the HOMO and LUMO levels of RhB shift when they are adsorbed to the surface of FICO as there is evidence of this happening with other molecules. How confident are the authors that the literature values for the placement of these bands in

solution reflect their values when adhered to FICO nanoparticles? Presumably, extending the Fowler plot in figure 4c would allow the authors to better assess the LUMO level position of RhB molecules when adhered to FICO.

Response: We thank reviewer raising this interesting question and the suggestion.

1) We expect no strong hybridization between FICO and RhB based on the absorption spectra. Strong hybridization will significantly damp the plasmon resonance and broaden the plasmon peak, as shown in previous study.¹² As shown in Fig. 1a or below, the plasmon peak is changed by less than 10% after RhB binding. We also compared the absorption spectrum of RhB in FICO-RhB complex and in methanol solution. We redshifted the wavelength of RhB solution by 7 nm because of different solvent polarity. The RhB peak width in FICO-RhB and RhB solution is also quite similar. These results suggest no strong hybridization between FICO and RhB. The absence of strong hybridization is also supported by indirect transfer mechanism instead of direct transfer.

2) following reviewer's suggestions, we have extended our excitation energy to even lower energy (0.605 eV or 1950 nm, 0.636 eV or 2050 nm). The quantum yield (QY) as a function of excitation energy are shown in Fig. 4b. The QY decreases monotonically with decreasing energy and shows a knee-behavior at ~ 0.68 eV. Below 0.67 eV, there is still measurable QY but the value is very small, which is likely due to sample heterogeneity. The results above 0.68 eV can be very well modeled assuming only the percentage of electrons with energy above barrier height after Landau damping can transfer to RhB LUMO (Fig. 4c). Based on the modeling, the barrier height is ~ 0.68 eV. The theoretical barrier height should be -3.94 (LUMO of RhB) - (-4.62) (Ef of FICO) = 0.68 eV. The E_b determined from pump energy dependent study is same as the theoretical value, which confirms the position of RhB LUMO level.

Revisions: in main content, We have revised Fig. 4b with results from lower photon energies added.

in main content, page 13 bottom, we revised

The electron percentage (γ) above E_b after Landau damping is determined by photon energy and E_b (Fig. 4c). In principle, no electrons from Landau damping can exceed E_b if photon energy is less than that, which sets a threshold value. Above that threshold, QY should increase with photon energy and the slope of increase is determined by transfer efficiency η . In Figure 4b, we show the modeled QY with different E_b values and E_b of 0.68 eV provides the best agreement with experimental results. The extracted 0.68 eV barrier height is same as the energy difference between estimated E_f (-4.62 eV) and RhB LUMO (-3.94 eV), confirming the proposed picture.

4. How much information do the authors have regarding the amount of RhB that binds to each nanoparticle? How are RhB molecules adhered to the FICO surface and is there expected to be a preferred geometry with which they bind? I ask as it would be helpful to know the distance scale over which electron transfer from FICO to RhB needs to take place and the density of acceptor molecules held at that distance to make electron transfer competitive with electron cooling.

Response: We thank reviewer for this interesting question. We estimated the average number of RhB molecules per FICO NC of ~ 90 , based on their extinction coefficients (SI note 2). According to previous studies on dye sensitized semiconductor oxides and NC ligand chemistry, we believe RhB molecules attach to FICO NCs through carboxylic group.¹⁴ As to the specific orientation or molecule configuration on NC surface (e.g. standing up vs laying down), we currently have no idea on that. They should form a distribution and we measured the averaged effect. Molecules laying down on FICO surface should have stronger coupling thus faster ET rate.

Revisions:

1) in SI 2, we added details of estimating average number of RhB per NC.

2) in main text, page 3, top, we added

We estimated the average number of RhB molecules per FICO NC to be ~ 90 based on their extinction coefficients (Supplementary Note 2).

5. Some additional details regarding the transient absorption layout used by the authors is warranted. The authors report use of a Pharos system, but what is the repetition rate used for transient absorption experiments? Likewise, what detector was used for reading out data? The authors also quote their instrument response function as being 290 fs, yet highlight time dynamics and spectral shifting in the main text at time delays shorter than this in figures 2, 3, and 4, such as a 50 fs rise of RhB bleaching. What is the reason the authors feel confident that such dynamics are simply not an experimental artifact tied to other nonlinear processes that can occur when the pump and probe pulses are temporally overlapped in the sample?

Response: 1) we are sorry we didn't describe our TA setup clearly. We have revised the methods part with all details of our setup.

2) We are sorry that 290 fs IRF in the supporting information was a typo, should be 200 fs as was specified in the main content. We determined the instrument response function using solvent response and the IRF is ~ 200 fs. For a 200 fs IRF, because we perform convolution fitting, the time resolution we can resolve is $IRF/4 \sim 50$ fs. That is to say, any process faster than 50 fs, we cannot extract it reliable anymore and they all just show up like instantaneously. But any process slower than 50 fs, we can still extract it reliable by convolution fitting. For electron transfer to RhB, the rising process is fitted to be < 50 fs and beyond our time resolution. We simply denote the process < 50 fs and cannot specify the exact value. For all other kinetics process, although the lifetime is on same magnitude as IRF, we can still reliably extract it by convolution fitting.

3) We performed additional TA experiment by tuning excitation wavelength over a broad range, including 0.605 eV or 1950 nm, 0.636 eV or 2050 nm and 1.55 eV or 800 nm. When we further reduce the excitation energy below Eb barrier height (~ 0.67 eV) but still in resonance with plasmon absorption, the RhB GSB signal is very small, approaching zero. When we tune excitation wavelength to 800 nm where no absorption from FICO and RhB, we observe no TA signal. All these indicate no coherent artifact from temporal overlap between pump and probe.

Revisions:

1) We have revised the Methods part with all details of our TA setup, indicating IRF and time resolution.

2) main content, page 9, middle, we added

The lifetime of rise and decay process of RhB bleach kinetics are < 50 fs (beyond time resolution) and 407 fs, respectively, corresponding to electron transfer to RhB and subsequent back electron transfer process.

main content, page 12, middle, we added

We also tuned excitation wavelength to 800 nm which is not in resonance with FICO and RhB transitions and observed no TA signal, precluding coherent artifacts from temporally overlapped pump and probe at experiment conditions.

6. The authors note a delayed rise of the data at 550 nm that they attribute to electron thermalization within the FICO conduction band. Showing data that fully highlights the development of this thermalized distribution (such as a contour map of the full spectral dynamics of the TA data) would be useful as well as a fit to the data from the authors' spectral model for electron thermalization.

Response: 1) In the original submission, our probe is limited to 500 nm. Now we can probe down to 400 nm therefore we can see clearly the evolution of thermalization process. The initial induced absorption right after Landau damping starts at 450 nm, which correspond to Fig. 2c left panel. During electron thermalization process through e-e and e-ph scattering, the IA onset shifts to lower energy until it reaches the bandgap B'_g (1.86 eV or 667 nm), as shown in Fig. 2c right panel. We show TA spectra in Fig. 2a for every 0.1 ps and show the 2D color plot in Fig. S3.

2) During the thermalization process, we could not fit the IA spectrum with Fermi-Dirac distribution as electrons are non-thermalized. We can only fit the spectrum after thermalization (at ~ 0.5 ps) with a well-defined electron temperature (4500 K).

Revisions: 1. we provided 2D color plot of all TA spectra in SI Fig. 3

2. in main content, we revised Fig 2a by adding 400-500 nm wavelength region and show TA spectra by 0.1 ps interval so that the spectral evolution can be clearly observed. We also provided associated discussions in main content.

7. In figure 4c, is the efficiency plotted an internal quantum efficiency that accounts for the FICO nanoparticle absorption strength at each probe energy or is it an external quantum efficiency that does not? I am not sure if the Fowler equation considers spectral variations in absorption strength, which could explain the differences in curvature with changing photon energy between the Fowler equation prediction and that shown by the experimental data. Also, as I mentioned in point 3, it would be good to extend this plot over a wider photon excitation energy range to demonstrably show that electron transfer fully turns off once the photon

energy is reduced below the threshold for electron transfer set by the energy difference between the FICO Fermi energy and RhB LUMO level.

Response: 1) The quantum yield plotted or discussed in manuscript is purely absorbed photon to injected electron efficiency that has corrected the plasmon absorption strength. In the revised manuscript, we stick to our own model (which basically same as Fowler model but with more insights and quantitative information on each steps). The modeled quantum yield is just a percentage which is not influenced by absorption profile.

2) Following reviewer's suggestion, we performed additional measurements with lower photon energies (0.605 eV or 1950 nm, 0.636 eV or 2050 nm). For photon energies below a certain threshold values E_b (~ 0.67 eV), QY decreases to almost zero but not absolutely zero (Fig. 4b). This can be attributed to sample heterogeneity especially doping level or fermi level heterogeneity.

Revisions: in main content, we have revised Fig. 4b with results from lower photon energies added.

in main content, page 13 bottom, we revised

The electron percentage (γ) above E_b after Landau damping is determined by photon energy and E_b (Fig. 4c). In principle, no electrons from Landau damping can exceed E_b if photon energy is less than that, which sets a threshold value. Above that threshold, QY should increase with photon energy and the slope of increase is determined by transfer efficiency η . In Figure 4b, we show the modeled QY with different E_b values and E_b of 0.68 eV provides the best agreement with experiments results. The extracted 0.68 eV barrier height is same as the energy difference between estimated E_f (-4.62 eV) and RhB LUMO (-3.94 eV), confirming the proposed picture.

8. What is the difference between the underlying assumptions made by the Fowler model and those used to produce the blue trace in figure 4c? I think the assumptions are very similar, so I'm not sure what the benefit is for showing both models.

Response: We thank reviewer raising this important point. Following reviewer suggestions, we looked into the details of Fowler model. Their key assumptions are similar but the model we proposed better fits to our study here.

Fowler model depicts the transfer efficiency across the Schottky junction from metal into semiconductor. The efficiency γ is calculated as:

$$\gamma = \frac{1}{2n} \int_0^{\pi/2} d\theta \sin\theta \int_0^{E_f} d\varepsilon \rho(\varepsilon) \Theta[(\varepsilon + \hbar\omega) \cos^2\theta - (E_f + eE_b)]$$

ε is the initial energy of the electron. Θ is the Heaviside unit-step function. θ is an angle with respect to the normal to the junction plane. $\rho(\varepsilon)$ is the density of states, and the DOS at Fermi surface is used for all the ε , i.e. $\rho(\varepsilon) \approx \rho(E_f)$. n is the total number of hot electron that can be excited:

$$n = \int_{E_f - \hbar\omega}^{E_f} \rho(\varepsilon) d\varepsilon$$

From the equation, we can see the calculation only consider the energy $(\varepsilon + \hbar\omega) \cos^2\theta$ of the motion in the direction normal to the junction plane with energy higher than the Schottky barrier height eE_b , and all the possible injection angles are integrated. This reduce to the Fowler equation, known as:

$$\gamma = \frac{(\hbar\omega - eE_b)^2}{eE_f \hbar\omega}$$

In the system where electron linear momentum is relaxed due the roughness of the junction, the γ is calculated as:

$$\gamma = \frac{1}{2n} \int_0^{\pi/2} d\theta \sin\theta \int_0^{E_f} d\varepsilon \rho(\varepsilon) \Theta[(\varepsilon + \hbar\omega) - (E_f + eE_b)]$$

This reduce to

$$\gamma = \frac{\hbar\omega - eE_b}{2\hbar\omega}$$

Our experimental result also can be fitted using non-conservation momentum Fowler equation.

Our own model, however, is similar but modified to fit our system better compared to the Fowler model.

$$\gamma = \frac{\int_{E_f - (\hbar\omega - E_b)}^{E_f} \rho(\varepsilon) \rho(\varepsilon + \hbar\omega) d\varepsilon}{\int_{E_f - \hbar\omega}^{E_f} \rho(\varepsilon) \rho(\varepsilon + \hbar\omega) d\varepsilon}$$

We only integrated the hot electron higher than the barrier height and introduced a transfer efficiency parameter. The main difference our model and Fowler equation are as follows:

1. In derivation of classical Fowler equation, hot electrons with different injection angles to the junction surface are integrated. Since the interface of our system is not a conventional semiconductor junction interface, we do not need to consider the injection angle. Instead, we introduce a transfer efficiency parameter in our own model to account for the transfer process.
2. The density of states $\rho(\varepsilon)$ in Fowler equation's derivation is using the DOS at Fermi surface, i.e. $\rho(\varepsilon) \approx \rho(E_F)$. This assumption is solid when the Fermi surface is much higher than the conduction band minimum (E_{CBM}) thus the DOS can be regarded as constant. In our system, the Fermi level is only 1.63 eV higher than E_{CBM} , thus we are using the parabolic assumption for the conduction band dispersion to calculate the DOS. It fits into our system better.

Another advantages of using our own model is we can have a clear physical picture and identify the efficiency limiting factors, e.g. low transfer efficiency in our system.

Revisions: 1) we removed Fowler model in Figure 4 and associated discussion in the main content. Instead we stick to our own model to model the experiment results and identify efficiencies in each step and perform analysis.

2) in SI 7 where we show the details of our own model, we also compare with Fowler model and discuss their difference. We also show the fit of using Fowler model in Fig. S6.

9. In section S3 of the supporting information, why is $T_{e,i}$ referenced as the "ideal" electron temperature? Why is it "ideal"? Is there a difference between your calculated $T_{e,i}$ and the $T_{e,i}$ you get by fitting the TA data?

Response: we are sorry we didn't describe it clearly in the previous submission. To make it easier to understand, in the revised manuscript, we denote it as theoretical maximum electron temperature $T_{e,max}$. As described in S3, it is the electron temperature assuming all excitation power is transferred and conserved in electron system in the conduction band without any energy loss to phonon. This approximation is based on faster electron-electron scattering than

electron-phonon scattering. The extracted T_e by fitting IA spectra is the experimental temperature after thermalization. This value is smaller than $T_{e,max}$ because of a small portion of energy loss from electron-phonon scattering during thermalization process.

Revisions: we revised S4 and in main content, page 7, top, we revised

The theoretical maximum electron temperature $T_{e,max}$ assuming all absorbed photon energies are transferred to conduction band electrons without any energy loss from phonon emission should be 5763 K. The extracted lower T_e for thermalized electrons indicates the onset of e-ph scattering during electron thermalization.

1. Manjavacas A, Liu JG, Kulkarni V, Nordlander P. Plasmon-induced hot carriers in metallic nanoparticles. *ACS Nano* 2014, **8**(8): 7630-7638.
2. Beaumont PC, Johnson DG, Parsons BJ. Excited state and free radical properties of rhodamine dyes in aqueous solution: A laser flash photolysis and pulse radiolysis study. *Journal of Photochemistry and Photobiology A: Chemistry* 1997, **107**(1-3): 175-183.
3. Boulesbaa A, Issac A, Stockwell D, Huang Z, Huang J, Guo J, *et al.* Ultrafast charge separation at CdS quantum dot/rhodamine B molecule interface. *J Am Chem Soc* 2007, **129**(49): 15132-+.
4. Sakamoto M, Kawawaki T, Kimura M, Vequizo JJM, Matsunaga H, Ranasinghe CSK, *et al.* Clear and transparent nanocrystals for infrared-responsive carrier transfer. *Nat Commun* 2019, **10**(1): 406.
5. Lian Z, Sakamoto M, Vequizo JJM, Ranasinghe CSK, Yamakata A, Nagai T, *et al.* Plasmonic p-n Junction for Infrared Light to Chemical Energy Conversion. *J Am Chem Soc* 2019, **141**(6): 2446-2450.
6. Lian Z, Sakamoto M, Matsunaga H, Vequizo JJM, Yamakata A, Haruta M, *et al.* Near infrared light induced plasmonic hot hole transfer at a nano-heterointerface. *Nature Communications* 2018, **9**(1): 2314.
7. Yu Y, Sun Y, Hu Z, An X, Zhou D, Zhou H, *et al.* Fast Photoelectric Conversion in the Near-Infrared Enabled by Plasmon-Induced Hot-Electron Transfer. *Adv Mater* 2019: e1903829.
8. Gan XY, Keller EL, Warkentin CL, Crawford SE, Frontiera RR, Millstone JE. Plasmon-Enhanced Chemical Conversion Using Copper Selenide Nanoparticles. *Nano Lett* 2019.

9. Christopher P, Moskovits M. Hot Charge Carrier Transmission from Plasmonic Nanostructures. *Annu Rev Phys Chem* 2017, **68**: 379-398.
10. Heilpern T, Manjare M, Govorov AO, Wiederrecht GP, Gray SK, Harutyunyan H. Determination of hot carrier energy distributions from inversion of ultrafast pump-probe reflectivity measurements. *Nat Commun* 2018, **9**(1): 1853.
11. Kriegel I, Urso C, Viola D, De Trizio L, Scotognella F, Cerullo G, *et al.* Ultrafast Photodoping and Plasmon Dynamics in Fluorine-Indium Codoped Cadmium Oxide Nanocrystals for All-Optical Signal Manipulation at Optical Communication Wavelengths. *J Phys Chem Lett* 2016, **7**(19): 3873-3881.
12. Wu K, Chen J, McBride JR, Lian T. Efficient hot-electron transfer by a plasmon-induced interfacial charge-transfer transition. *Science* 2015, **349**(6248): 632-635.
13. Asbury JB, Hao E, Wang YQ, Ghosh HN, Lian TQ. Ultrafast electron transfer dynamics from molecular adsorbates to semiconductor nanocrystalline thin films. *J Phys Chem B* 2001, **105**(20): 4545-4557.
14. Boles MA, Ling D, Hyeon T, Talapin DV. The surface science of nanocrystals. *Nat Mater* 2016, **15**(2): 141-153.

Reviewer #1 (Remarks to the Author):

I have found that the authors have revised the manuscript very carefully regarding my comments. All revised parts are reasonable and I would like to recommend this version for publication.

Reviewer #2 (Remarks to the Author):

I think the authors have addressed the comments satisfactorily. I recommend the manuscript for acceptance.

Reviewer #3 (Remarks to the Author):

In their revised the authors have provided two key control experiments I requested in my initial review of the manuscript: (1) definitive proof that charge transfer was occurring by identifying the photoinduced absorption signal of the rhodamine B anion, and (2) transient absorption data measured using pump photons that can excite the FICO LSPR but lack sufficient energy to drive hot electron transfer to rhodamine B. With inclusion of this data, I am now convinced the authors indeed observe hot electron transfer. As such, I do believe their work carries enough import to merit publication in Nature Communications.

However, there are still some items I find confusing regarding the authors' discussion of their data. In particular, some aspects regarding how the authors account for the initial temperature of the system, photon absorption statistics, and general fitting of their data are still a bit puzzling. With improvements to these issues outlined below, I believe this manuscript will be suitable for publication in Nature Communications:

- 1) I am confused regarding how the authors account for the photon statistics of light absorption within their model. Specifically, I am referring to the discussion surrounding figure 4a, which shows the ground state bleach intensity for rhodamine B scales linearly with excitation pump fluence. This is exactly the result one would expect if the authors are operating in a low fluence regime where each nanocrystal in their sample has a probability of absorbing either one or zero photons. Each

photon would impart an equal amount of energy to a nanocrystal and the linear scaling would result as the number of excited nanocrystals would scale linearly with the photon flux (i.e. fluence). It is only at fluences wherein the probability of absorbing multiple photons within a single nanocrystal becomes significant that one would expect to start to see a fluence dependence to the hot carrier distribution produced within photoexcited nanocrystals.

It is not clear from the text if the authors' model used to calculate the "thermalized" curve accounts for the statistics of individual nanocrystals absorbing zero, one, or multiple photons, which is presumably governed by a Poisson distribution. If not, then their model is effectively assuming that all nanocrystals within their sample act to thermalize the total amount of energy absorbed, which is simply not the case as the nanocrystals are not in electrical contact. I suspect that if the authors have not accounted for these statistics, the experimental result in figure 4a may simply be a trivial one that provides no information regarding the mechanism of electron transfer to rhodamine B.

2) Looking at the equation given in Supplementary Note 7 for calculating gamma, the percentage of electrons with energy above the LUMO energy of rhodamine B, I am a little puzzled as this seems to neglect the initial temperature of the system. This appears to be a zero-temperature expression where the authors have assumed all states below the Fermi energy are filled and all states above it are empty. However, a more exact accounting would be to convolve the density of filled conduction band states at the temperature at which experiments are performed (presumably room temperature) with the density of empty states. I suspect this may increase the value of gamma, which would lower the authors electron transfer efficiency estimate of 5.5%.

3) I applaud the authors for making additions to their transient absorption spectrometer that enabled them to record spectra of the rhodamine B anion, which clearly shows electron transfer from FICO to rhodamine B. However, I am puzzled that the authors then did not focus on the kinetics shown by that band for their data analysis. The criticisms that I raised in my prior review regarding issues tied to use of the ground state bleach for analysis of electron transfer kinetics still stand. Portions of the rise and fall of the ground state bleach of rhodamine B could be subject to other effects, such as a photoinduced stark effect or plasmon heating of the nanocrystal following LSPR relaxation. The anion band on the other hand should provide a much clearer measure of any FICO to rhodamine B electron transfer dynamics, which I would think would make it superior for extracting time constants for this process.

I acknowledge the authors stated their data showing the appearance of the anion band exhibits worse signal to noise than their data showing the ground state bleach of rhodamine B, which is why

they have continued to use the latter for kinetics analysis. However, the authors should at the very least show a plot indicating the temporal dynamics of the rise and fall of the rhodamine B anion radical are consistent with those shown by the rhodamine B ground state bleach if they insist on using the bleach for extracting electron transfer/recombination time constants.

4) In my prior review, I asked that the authors include some additional data regarding the laser system they use for transient absorption measurements, with a specific emphasis placed on how they determine their time resolution. The authors have revised the description of their instrument response function, noting that it was 200 fs long rather than 290 fs as quoted in their original manuscript.

First, the authors should include some data in the Supporting Information showing this is indeed the case. My experience with Yb:KGW fiber lasers such as the Pharos used by the authors is that they tend to produce pulses that are much closer to 300 fs in length rather than 200. Frequency conversion of the laser fundamental to produce pump pulses for these experiments should only broaden the pulse more, not shorten it.

Second, the authors state they can measure time constants that are up to 4 times shorter (~ 50 fs) than their stated instrument response. I find this a bit shocking and would really like to see some fits to their experimental data that employ different decay rates that are within their stated instrument response function. From personal experience, deconvolving an instrument response function to obtain a shorter time constant is hard to do accurately and subject to large error. Getting an accurate measure of the electron transfer timescale is important as it speaks to the authors' arguments regarding if electron transfer occurs prior to or following electron thermalization.

5) No equations are given that describe the kinetic models used to fit either the FICO cooling data shown in figure 2b or the electron transfer/recombination kinetics in figure 3c. To fit the FICO cooling data, are the authors using the canonical two temperature model that has been applied to plasmonic nanoparticles? If so, is their extracted cooling rate of 210 fs consistent with reports of the electron-phonon coupling constant of FICO?

I also find it strange that the authors obtain a cooling time of 210 fs when their data shows a maximum in figure 2b that peaks close to 500 fs. Playing around with simple kinetic models involving intermediate states, I was not able to reproduce this result. The authors need to clarify their model.

6) I thank the authors for computing the number of rhodamine B molecules that adhere to their particles, which I requested in my prior review, but they really should discuss the influence that the number of bound molecules has on their measured electron transfer rate. In a naïve picture, if each rhodamine B molecule bound to a nanocrystal is equally capable of accepting a hot electron, this means the effective electron transfer rate measured should be scaled down by the number of acceptors (i.e. the intrinsic electron transfer timescale from FICO to rhodamine B is $\sim(50 \text{ fs}) * 90 = 4.5 \text{ ps}$).

7) The language in the article could use some proof-reading. While I was able to follow most of the authors' discussion, there are a few odd metaphors, "knee behavior" for example, that I had to read a few times before following the authors' discussion. There are also some missing/misplaced articles that if addressed could aid the manuscript's general readability.

We sincerely thank the careful reading and constructive feedback by 3rd reviewer, which help us significantly improve this manuscript. Following the reviewer's additional questions and comments, we've made the following corrections and improvements point by point in response. **The responses are in red** and **revisions are in blue**. The manuscript is revised using track-change mode and the marked version is submitted for review. The page number is referred to marked version.

Reviewer #3 (Remarks to the Author):

In their revised the authors have provided two key control experiments I requested in my initial review of the manuscript: (1) definitive proof that charge transfer was occurring by identifying the photoinduced absorption signal of the rhodamine B anion, and (2) transient absorption data measured using pump photons that can excite the FICO LSPR but lack sufficient energy to drive hot electron transfer to rhodamine B. With inclusion of this data, I am now convinced the authors indeed observe hot electron transfer. As such, I do believe their work carries enough import to merit publication in Nature Communications.

We sincerely thank reviewer for the kind suggestion and the positive comments.

However, there are still some items I find confusing regarding the authors' discussion of their data. In particular, some aspects regarding how the authors account for the initial temperature of the system, photon absorption statistics, and general fitting of their data are still a bit puzzling. With improvements to these issues outlined below, I believe this manuscript will be suitable for publication in Nature Communications:

We have made following revisions and improvements as shown below and hope the reviewer can be satisfied.

1) I am confused regarding how the authors account for the photon statistics of light absorption within their model. Specifically, I am referring to the discussion surrounding figure 4a, which shows the ground state bleach intensity for rhodamine B scales linearly with excitation pump fluence. This is exactly the result one would expect if the authors are operating in a low fluence regime where each nanocrystal in their sample has a probability of absorbing either one or zero photons. Each photon would impart an equal amount of energy to a nanocrystal and the linear scaling would result as the number of excited nanocrystals would scale linearly with the photon flux (i.e. fluence). It is only at fluences wherein the probability of absorbing multiple photons within a single nanocrystal becomes significant that one would expect to start to see a fluence dependence to the hot carrier distribution produced within photoexcited nanocrystals.

It is not clear from the text if the authors' model used to calculate the "thermalized" curve accounts for the statistics of individual nanocrystals absorbing zero, one, or multiple photons, which is presumably governed by a Poisson distribution. If not, then their model is effectively

assuming that all nanocrystals within their sample act to thermalize the total amount of energy absorbed, which is simply not the case as the nanocrystals are not in electrical contact. I suspect that if the authors have not accounted for these statistics, the experimental result in figure 4a may simply be a trivial one that provides no information regarding the mechanism of electron transfer to rhodamine B.

Response: we thank reviewer raising this interesting question. As reviewer pointed out, if the average photon number $\langle N \rangle$ absorbed per NC is less than 1, a linear relationship will always be obtained as we simply increase the excited NC population. In that way, we cannot obtain information about the mechanism. In the revised manuscript, we calculated the average number of photons absorbed per NC and this value in our experiment is much larger than 1. This is reasonable as this NC size is large and plasmon absorption cross-section is large. Therefore, when we change the pump fluence, we are indeed changing the number of hot electrons in each NC, not the percentage of photoexcited NCs. With this, we can say this linear relationship indicates electron transfer before electron-electron thermalization. Thanks for pointing it out.

Revisions:

1) we added Supplementary Note 7 where we show the calculation details of averaged number of photons absorbed per NC.

2) in Fig. 4a, we added $\langle N \rangle$ value as top axis,

3) in main content, page 11, bottom, we added

It is important to note we calculated the averaged photon number absorbed per NC $\langle N \rangle$ (Supplementary Note 7) and this value is much larger than 1 (Fig. 4a). Therefore, the linear power dependence is not due to linearly increased population of photoexcited NC.

2) Looking at the equation given in Supplementary Note 7 for calculating gamma, the percentage of electrons with energy above the LUMO energy of rhodamine B, I am a little

puzzled as this seems to neglect the initial temperature of the system. This appears to be a zero-temperature expression where the authors have assumed all states below the Fermi energy are filled and all states above it are empty. However, a more exact accounting would be to convolve the density of filled conduction band states at the temperature at which experiments are performed (presumably room temperature) with the density of empty states. I suspect this may increase the value of gamma, which would lower the authors electron transfer efficiency estimate of 5.5%.

Response: We appreciate reviewer for this great advice! In our initial modeling, we indeed neglect the actual temperature (RT) of the system and simply use 0 K. A more realistic one should be 298 K. In the revised manuscript we improve the model by using the room temperature distribution for both occupied states and empty states and integrating all the possible occupied states and empty states. The improved model has negligible influence on the transfer efficiency (this is because the RT thermal distribution is small compare to excited portion by high photon energy and transfer efficiency is determined from high energy linear region) but has strong impact for the lower energy photon near threshold as shown in Fig. 4a. The tail contribution can be captured much better with reviewer's advice.

Revisions:

1) we have revised SI7 and use FD distribution at 298 K.

2) We have changed the modeled quantum yield plot in Fig. 4a with updated model in SI7, as shown below.

3) main content, page 12, middle, we added,

At room temperature (298 K), the threshold would be obscured by thermal distribution of electrons near fermi level and there would be a minor contribution extending below threshold. This is exactly what we observed experimentally.

3) I applaud the authors for making additions to their transient absorption spectrometer that enabled them to record spectra of the rhodamine B anion, which clearly shows electron transfer from FICO to rhodamine B. However, I am puzzled that the authors then did not focus on the kinetics shown by that band for their data analysis. The criticisms that I raised in my prior review regarding issues tied to use of the ground state bleach for analysis of electron transfer kinetics still stand. Portions of the rise and fall of the ground state bleach of rhodamine B could be subject to other effects, such as a photoinduced stark effect or plasmon heating of the nanocrystal following LSPR relaxation. The anion band on the other hand should provide a much clearer measure of any FICO to rhodamine B electron transfer dynamics, which I would think would make it superior for extracting time constants for this process.

I acknowledge the authors stated their data showing the appearance of the anion band exhibits worse signal to noise than their data showing the ground state bleach of rhodamine B, which is why they have continued to use the latter for kinetics analysis. However, the authors should at the very least show a plot indicating the temporal dynamics of the rise and fall of the rhodamine B anion radical are consistent with those shown by the rhodamine B ground state bleach if they insist on using the bleach for extracting electron transfer/recombination time constants.

Response: we thank reviewer raising the issue of RhB anion kinetics and the understanding.

As shown in SI Fig5, the small RhB anion signal is overlapped with a much stronger IA signal from FICO while the signal from FICO has little contribution on RhB GSB. Therefore, in the kinetics and signal size analysis (for mechanism and quantum yield) which requires quantitative analysis with high quality data, we use RhB GSB. We appreciate reviewer for the understanding.

Following reviewer's suggestions, in the revised submission, we plotted the RhB anion kinetics with RhB GSB kinetics. The RhB anion generally follow the kinetics of GSB but with a large noise and fluctuation due to overlap with strong IA signal from FICO.

Revisions: we added the comparison between RhB anion and GSB kinetics in SI Fig.5 following reviewer's suggestion.

4) In my prior review, I asked that the authors include some additional data regarding the laser system they use for transient absorption measurements, with a specific emphasis placed on how they determine their time resolution. The authors have revised the description of their instrument response function, noting that it was 200 fs long rather than 290 fs as quoted in their original manuscript.

First, the authors should include some data in the Supporting Information showing this is indeed the case. My experience with Yb:KGW fiber lasers such as the Pharos used by the authors is that they tend to produce pulses that are much closer to 300 fs in length rather than 200. Frequency conversion of the laser fundamental to produce pump pulses for these experiments should only broaden the pulse more, not shorten it.

Response: The first amplification stage (preamp) of the OPA we use is noncollinear, which should be able to provide a short laser pulse. Following reviewer’s suggestion, we have shown the measured pump-probe cross correlation (as IRF) with 1500 nm excitation and at 560 nm probe wavelength. We fit the IRF with Gaussian function with a FWHM \sim 0.2 ps.

Revisions: in SI, we added Fig. S7 where we show the IRF function and the fit.

In main content, page 15, middle, we revised

“was determined from solvent response to be 200 fs (Supplementary Figure 7)”

Second, the authors state they can measure time constants that are up to 4 times shorter (~50 fs) than their stated instrument response. I find this a bit shocking and would really like to see some fits to their experimental data that employ different decay rates that are within their stated instrument response function. From personal experience, deconvolving an instrument response function to obtain a shorter time constant is hard to do accurately and subject to large error. Getting an accurate measure of the electron transfer timescale is important as it speaks to the authors' arguments regarding if electron transfer occurs prior to or following electron thermalization.

Response: We thank reviewer raising this issue. For IRF with FWHM of ~ 200 fs, the shortest lifetime we can extract is ~ 200/4=50 fs. Below that, we can only say < 50 fs. We illustrate this point by the following simulation. We perform a convolution between a IRF Gaussian function of FWHM = 200 fs with a single exponential rising process of different lifetimes. The convoluted kinetics are shown below. The instantaneous process (e.g. 10 fs) with a 200 fs IRF is shown in yellow. The rising process with 25 fs (red) almost overlaps with instantaneous one, which means we cannot resolve it with 200 fs IRF. But for the process with 50 fs lifetime, it's separated from instantaneous one and can be distinguished. Therefore, practically, this can be considered as the limit we can extract through convolution fitting. For 100 fs or 200 fs which is shorter than IRF, we can extract them reliably with convolution fitting.

In our experiment, RhB GSB grows as the yellow or red kinetics with almost instantaneous rising. That means it's faster than 50 fs. Therefore, in this study, we cannot provide any accurate lifetime number but we simply point out < 50 fs.

5) No equations are given that describe the kinetic models used to fit either the FICO cooling data shown in figure 2b or the electron transfer/recombination kinetics in figure 3c. To fit the FICO cooling data, are the authors using the canonical two temperature model that has been applied to plasmonic nanoparticles? If so, is their extracted cooling rate of 210 fs consistent with reports of the electron-phonon coupling constant of FICO?

Response: as we stated in manuscript, the kinetics in Fig. 2b and 3c were both fitted with an exponential rise and decay function convoluted with IRF, without any physical modeling. The physical model has been provided based on the spectral evolution. We simply extract the lifetime constant of these processes with this exponential fitting. Following reviewer suggestion, we provided the expression for the kinetics fitting.

Revisions: in main content,

page 8, middle, we revised as

We fit the kinetics with an exponential rise and decay function convoluted with IRF, i.e. $\Delta T/T(t) = IRF \otimes (-e^{-t/\tau_r} + e^{-t/\tau_D})$ where τ_r and τ_D is the rising and decay lifetime constant, respectively.

page 9, bottom, we revised as

we fit it with an exponential rise and decay function convoluted with IRF, i.e. $\Delta T/T(t) = IRF \otimes (-e^{-t/\tau_r} + e^{-t/\tau_D})$.

I also find it strange that the authors obtain a cooling time of 210 fs when their data shows a maximum in figure 2b that peaks close to 500 fs. Playing around with simple kinetic models involving intermediate states, I was not able to reproduce this result. The authors need to clarify their model.

Response: As stated above, we simply use conventional exponential rise and decay to fit the kinetics without any intermediate states. In exponential function, the lifetime obtained is not the time to reach signal maximum. Also, the convolution with IRF elongates the rising process. As shown in simulated kinetic curves above for Q4, a 200 fs rising lifetime reaches maximum at ~ 0.6 ps.

6) I thank the authors for computing the number of rhodamine B molecules that adhere to their particles, which I requested in my prior review, but they really should discuss the influence that the number of bound molecules has on their measured electron transfer rate. In a naïve picture, if each rhodamine B molecule bound to a nanocrystal is equally capable of accepting a hot electron, this means the effective electron transfer rate measured should be scaled down by

the number of acceptors (i.e. the intrinsic electron transfer timescale from FICO to rhodamine B is $\sim(50 \text{ fs}) \cdot 90 = 4.5 \text{ ps}$).

Response: we thank reviewer for this interesting thought! After careful thinking, we note this could be a complicated question. Unlike in quantum confined semiconductor nanocrystals e.g. quantum dots where electron wavefunction is delocalized over nanoparticles, the FICO NC in this study is fairly large and not quantum confined. Therefore, hot electrons generated in different locations should only interact with electron acceptors nearby. We also not clear where the doped electron is in the NC (inside or outside). So it's quite difficult for us to discuss the ET rate as function of Rhb numbers. As we discussed in the manuscript, some electrons might need to transport to surface then transfer to RhB, which causes the low transfer efficiency. With lowering RhB molecules, these electrons might not be able to transfer anymore. So overall, this system is not a simple point-like donor with a point-like acceptor which can be described by a simple kinetics model. At current stage, we feel difficult and unsafe to discuss amount of RhB on ET rate. Intuitively, for any reason, lowering RhB molecules will lower the transfer efficiency.

7) The language in the article could use some proof-reading. While I was able to follow most of the authors' discussion, there are a few odd metaphors, "knee behavior" for example, that I had to read a few times before following the authors' discussion. There are also some missing/misplaced articles that if addressed could aid the manuscript's general readability.

Response: we thank reviewer for the careful reading and kind suggestion. We have gone through the manuscript carefully and made corrections.

In their latest revision to their manuscript, the authors have addressed most of my concerns. However, there are two minor points in their reply that should be clarified prior to publication that pertain to comments 4 and 5 in my last review. Further review is not needed.

- 1) In the authors' reply to comment 4, they mentioned that a noncollinear optical parametric amplifier (NOPA) was used rather than a standard optical parametric amplifier for pulse generation. This should be explicitly stated in the manuscript as NOPAs generally produce shorter pulses than their input while OPAs do not.
- 2) I still find it strange that the ground state bleach shown in figure 3c peaks so late (~ 400 fs) given the timescales listed for electron thermalization and recombination (150 fs and 210 fs, respectively). The authors state that they fit their data with a function consisting of their instrument response (200 fs FWHM) convolved with the difference of an exponential rise and a decay.

I've tried to reproduce their fit below. Plotted in blue is the difference of two exponential functions with $\tau_r = 150$ fs and $\tau_D = 210$ fs. As expected given the timescales, the function peaks somewhere between the two rate constants, at ~ 180 fs. Convolution with a 200 fs instrument response (red curve) pushes the maximum away to a slightly longer point in time that is closer to 200 fs. The authors' data peaks much later than this though, closer to 400 fs. It is also strange that the authors' data does not contain show any significant signal at times prior to $t = 0$. Convolution with a function that rises from zero with a finite instrument response function should yield a pre-time zero component as can be seen in the red trace.

Rather, I can obtain a trace that approximates the author's data in figure 3c if I arbitrarily shift the

time axis by 200 fs, which forces the convolved data to rise from zero (black trace). However, this isn't proper to do as it implies a delayed rise of the signal. As the authors are trying to extract electron cooling rates and charge recombination timescales that are on the order of their time resolution, making sure that their fitting procedure properly accounts for the data's time origin is important. If the authors have arbitrarily shifted the time axis, I would recommend either replotting their data or noting the shift they've applied to it in the figure caption.

In their latest revision to their manuscript, the authors have addressed most of my concerns. However, there are two minor points in their reply that should be clarified prior to publication that pertain to comments 4 and 5 in my last review. Further review is not needed.

- 1) In the authors' reply to comment 4, they mentioned that a noncollinear optical parametric amplifier (NOPA) was used rather than a standard optical parametric amplifier for pulse generation. This should be explicitly stated in the manuscript as NOPAs generally produce shorter pulses than their input while OPAs do not.

Response: we thank reviewer for the suggestion. We have explicitly specified the manufacture and model of the optical parametric amplifier we use in this study.

Revision: page 16, method part, we revised

“One was introduced to a optical parametric amplifier to generate pump pulse at a certain wavelength in NIR (Light Conversion, OPA Orpheus-One).”

- 2) I still find it strange that the ground state bleach shown in figure 3c peaks so late (~400 fs) given the timescales listed for electron thermalization and recombination (150 fs and 210 fs, respectively). The authors state that they fit their data with a function consisting of their instrument response (200 fs FWHM) convolved with the difference of an exponential rise and a decay.

I've tried to reproduce their fit below. Plotted in blue is the difference of two exponential functions with $\tau_r = 150$ fs and $\tau_D = 210$ fs. As expected given the timescales, the function peaks somewhere between the two rate constants, at ~180 fs. Convolution with a 200 fs instrument response (red curve) pushes the maximum away to a slightly longer point in time that is closer to 200 fs. The authors' data peaks much later than this though, closer to 400 fs. It is also strange that the authors' data does not contain show any significant signal at times prior to $t = 0$. Convolution of a function that rises from zero with a finite instrument response function should yield a pre-time zero component as can be seen in the red trace.

Rather, I can obtain a trace that approximates the author's data in figure 3c if I arbitrarily shift the time axis by 200 fs, which forces the convolved data to rise from zero (black trace). However, this isn't proper to do as it implies a delayed rise of the signal. As the authors are trying to extract electron cooling rates and charge recombination timescales that are on the order of their time resolution, making sure that their fitting procedure properly accounts for the data's time origin is important. If the authors have arbitrarily shifted the time axis, I would recommend either replotting their data or noting the shift they've applied to it in the figure caption.

Response: We thank reviewer for the careful reading. In TA measurement, we can't determine the absolute time zero position. Instead, the time zero position is captured by the fitting. The relative time zero doesn't affect the fitted lifetime. Following reviewer's suggestion, we have shifted the time axis according to fitted time zero so that the time zero is truly at 0.

Revision: We have shifted the relative time in Fig.2 and Fig.3 to make it easier to visualize.